# The climatology and nature of warm-season convective cells in cold-frontal environments over Germany

George Pacey[1], Stephan Pfahl[1], Lisa Schielicke[1,2], and Kathrin Wapler[3]

[1]Institute of Meteorology, Freie Unversität Berlin, Berlin, Germany
[2]Department of Meteorology, University of Bonn, Bonn, Germany
[3]Deutscher Wetterdienst, Offenbach, Germany

**Correspondence:** George Pacey (george.pacey@fu-berlin.de)

**Abstract.** Cold fronts provide an environment particularly favourable for convective initiation in the mid-latitudes and can also be associated with convective hazards such as flooding, wind, hail and lightning. We build a climatology of cold-frontal convective cells between 2007–2016 for April–September in a cell-front distance framework by combining a radar-based cell detection and tracking dataset and automatic front detection methods applied to reanalysis data. We find that on average around

twice as many cells develop on cold-frontal cell days compared to non-cold-frontal cell days. Using the 700 hPa level as a reference point we show the maximum cell frequency is 350–400 km ahead of the 700 hPa front, which is marginally ahead of the typical surface front location. The 700 hPa front location marks the minimum cell frequency and a clear shift in regime between cells with a weakened diurnal cycle on the warm-side of the 700 hPa cold front and strongly diurnally driven cells on the cold-side of the 700 hPa front. High cell frequency is found several hundreds of kilometres ahead of the surface front

and cells in this region are most likely to be associated with mesocyclones, intense convective cores and lightning. Namely, mesocyclones were detected in around 5.0% of pre-surface-frontal cells compared to only 1.5% of non-cold-frontal cells. The findings in this study are an important step towards a better understanding of cold-frontal convection climatology and links between cold fronts and convective hazards.

## 1 Introduction

Atmospheric convection is a key process in the formation of certain atmospheric hazards such as flooding, wind, hail and lightning. In Europe, convective loss events to the insurance industry have risen by around a factor of 4 from about 30 events per year in 1980 to about 120 in 2014 (Hoeppe, 2016). Germany is particularly prone to such convective loss events with some events leading to insurance losses of over 1 billion euros (Kunz et al., 2018; Wilhelm et al., 2021). Taszarek et al. (2019) showed based on European Severe Weather Database reports (ESWD; Dotzek et al., 2009) that much of Germany reports

8 or more days per year with heavy convective precipitation (their Figure 7). Recent events such as the exceptional floods in Central Europe in July 2021 (Mohr et al., 2023), which was exacerbated by embedded convection, highlight the need for further information on the underlying processes that lead to heavy convective precipitation.

Deep moist convection (DMC) frequently initiates in proximity to air mass boundaries such as synoptic-scale fronts (length scale of ~1000 km), drylines and outflow boundaries (Markowski and Richardson, 2010). While cold-frontal convection is

frequent during the warm-season, convection rarely initiates along the entirety of the boundary. This indicates that the precise location of convective initiation is likely due to underlying mesoscale processes (Markowski and Richardson, 2010). Such processes are not currently well-understood, thus forecasting of cold-frontal convection is challenging. Furthermore, the importance of different processes likely vary across the front, e.g, stronger forcing near the front but stronger solar heating away from the front. Previous studies including field experiments (e.g., Wulfmeyer et al., 2011, Bott, 2016a; Lee et al., 2016) have demonstrated that fronts can favour the occurrence of convection due to different processes depending on the convective initiation location relative to the front.

The region ahead of a cold front is typically referred to as pre-frontal, which typically corresponds to the warm sector of extratropical cyclones. Convection can be triggered by the convergence of warm and humid air masses near the surface. For example, near the surface front or at pre-surface-frontal convergence lines (e.g., Wulfmeyer et al., 2011; Dahl and Fischer, 2016). The ageostrophic frontal circulation leads to lifting (Bluestein, 1993) which can help air parcels overcome convective inhibition (CIN) as well as increase convective available potential energy (CAPE) through adiabatic cooling (Trapp, 2013; his Figure 5.2). The high temperatures and moisture availability pre-frontal also increase CAPE. Convection may also be embedded in stratiform rain regions, sometimes in the form of narrow cold-frontal rainbands (Gatzen, 2011; Clark, 2013). These mostly occur in the cool-season and thus will not be explicitly addressed in this study. Convection may also initiate behind a cold front in the post-frontal environment mainly due to the destabilisation of the air masses associated with the transport of colder air over a potentially warmer surface (Weusthoff and Hauf, 2008; Bott, 2016a). Convection can occur simultaneously pre-frontal, frontal and post-frontal depending on environmental conditions (Ferretti et al., 2014).

Since cold fronts typically slope backwards with height, there is no concrete and uniform definition of post-frontal, frontal and pre-frontal convection in the literature. Some studies consider any convection after the passage of the surface cold front to be post-frontal (e.g., Schumacher et al., 2022), including convection embedded in the frontal stratiform rainband. As a surface observer this holds true, but the saturated cloud region and thus any embedded convection is likely located on the warm-side of the cold front so could be considered frontal convection. Other studies consider post-frontal convection to be the diurnally driven mostly surface-based convection that is typically seen in the form of isolated showers or thunderstorms (e.g., Weusthoff and Hauf, 2008; Bott, 2016a). Where the transition from post-frontal to frontal and frontal to pre-frontal should be is also arbitrary. Therefore, we produce a cold-frontal convective cell climatology largely in a cell-front distance framework.

Previous studies on cold-frontal convection in Europe have primarily focused on narrow cold-frontal rainbands (e.g., Gatzen, 2011; Clark, 2013) or on cold-frontal hailstorms (Schemm et al., 2016; Kunz et al., 2020). Schemm et al. (2016) showed up to 45% of all detected hail events in north-eastern and southern Switzerland form in pre-frontal zones arguing that vertical wind shear and along frontal moisture transport were among the mechanisms favouring hail formation in pre-frontal environments. Kunz et al. (2020) found frontal convective storms associated with hail produce larger hailstones and have longer tracks on average and that a front was needed to trigger 50% of severe convective storms over flat-terrain in Germany. Large hail ($\geq 2$ cm diameter), especially very large hail ($\geq 5$ cm diameter), is often associated with supercells (e.g., Smith et al., 2012; Wapler, 2017). The defining characteristic of a supercell is a mesocyclone, that is, a deep and persistent rotating updraft. The frequency of mesocyclones and the associated hazards depending on the cell-front distance remain unexplored in any detail.

In this study we focus on warm-season cold-frontal convection (April–September) and use a broader definition of convection which is based on radar reflectivity and area thresholds thus is not solely based on convective lines (as in Gatzen, 2011 and Clark, 2013). We seek to shed light on the climatology and nature of convective cells in cold-frontal environments over Germany in a comprehensive framework, that is to say, considering the climatology and nature of cells depending on the distance from the front. By also considering lightning, mesocyclones and cell intensity in the same framework, we aim to improve understanding of convective hazard climatology in cold-frontal environments. To our knowledge, a study incorporating such aspects is not present in the current literature. We also seek to highlight the differences between cold-frontal convective cells and non-cold-frontal convective cells. The primary research questions addressed in this study are as follows:

Q1) How do the spatial and temporal frequency of convective cells differ depending on the cell's distance from the front?

Q2) How does the nature of convective cells differ depending on the cell's distance from the front?

Q3) How do Q1 and Q2 compare to non-cold-frontal convective cells?

The paper is organised as follows: Section 2 will introduce the convective cell detection and tracking dataset as well as the automatic front detection methods applied to ERA5 reanalysis data. We also show front detection examples for cases with convective cells in proximity. Section 3 will highlight the key results of the study and put them in the context of the current literature. The subsections focus on climatological cold-frontal environments, convective cell count and cell days, diurnal cycle, spatial climatology and the nature of convective cells. For the nature of cells we investigate cell lifetime, propagation speed, organisation, lightning frequency, cell intensity, and mesocyclone frequency. The nature of cells section will therefore provide novel insights into how convective hazard climatology may vary depending on the distance from the front. In Section 4 we summarise the results and highlight the importance of this study.

## 2 Data and Methods

Two datasets are combined: ERA5 reanalysis data (Hersbach et al., 2018, 2020) and the KONRAD Convective Cell Detection and Tracking Dataset (Wapler and James, 2014). Automatic front detection methods are applied to the ERA5 reanalysis dataset (Section 2.1) and KONRAD (Section 2.2) is used for the convective cell definitions. At the start of the study, ERA5 was available between 1959 to present whereas KONRAD was available from 2007–2016 for the months April–September, thus the KONRAD availability period was used. The spatial and temporal resolutions of ERA5 are 1 hour and 0.25 degrees and 5 minutes and 1 km for KONRAD.

### 2.1 Front Detection

A front is generally considered a boundary between air masses with different properties such as temperature and humidity. Forecasters are primarily interested in synoptic-scale fronts as they are the main driver of precipitation in the mid-latitudes (Catto and Pfahl, 2013). Forecasters often manually position such fronts based on numerical weather prediction (NWP) output

and observations. They sometimes use more technical algorithms. However, Renard and Clarke (1965) noted that different forecasters produce different analyses thus the final front location can be subjective depending on the forecaster. An additional problem is that archives of synoptic charts with fronts are sparse in both space and time. Renard and Clarke (1965) was one of the first studies that recognised the need for objective methods to detect fronts, and since then, automatic front detection in reanalysis data has attracted increasing attention (e.g., Hewson, 1998; Jenkner et al., 2010; Schemm et al., 2015, 2016, 2018; Thomas and Schultz, 2019; Rüdisühli et al., 2020). All of these studies reference the Thermal Front Parameter equation (Equation 1), which was first introduced by Renard and Clarke (1965).

$$\text{TFP} = -\nabla|\nabla\tau| \cdot \frac{\nabla\tau}{|\nabla\tau|} \tag{1}$$

The term represents the rate of change of $\tau$ projected in the direction of the thermal gradient, where $\tau$ is a thermodynamic variable (e.g., potential temperature or equivalent potential temperature). The projection takes the curvature of fronts into account. Since the term is a double derivative an inflexion point is found where the parameter is equal to zero. The maximum thermal gradient is thus where TFP=0. The horizontal wind ($\mathbf{v}$) can be projected onto the frontal line using Equation 2 (Hewson, 1998).

$$v_f = \mathbf{v} \cdot \frac{\nabla(\text{TFP})}{|\nabla(\text{TFP})|} \tag{2}$$

The term $v_f$ is the horizontal wind ($\mathbf{v}$) projected in the direction of the TFP gradient. It is positive at cold fronts and negative at warm fronts. The $v_f$ threshold can be altered to consider fronts meeting a minimum advection criteria. A higher magnitude of $v_f$ indicates stronger advection at the frontal boundary. Stationary fronts are found where $v_f \approx 0$.

### 2.1.1 Criteria

In this study, the cold-frontal regions are detected using the following criteria:

$$|\nabla\theta_e| > 6\,\text{K}(100\,\text{km})^{-1} \tag{A}$$

$$v_f > 1\,\text{m}\,\text{s}^{-1} \tag{B}$$

$$L > 1000\,\text{km} \tag{C}$$

The equivalent potential temperature is represented by $\theta_e$ and is used as the thermodynamic variable $\tau$. The along front length is represented by $L$. The overlap of the $\theta_e$ gradient threshold (condition A) and velocity threshold (condition B) represents the front contour. Since we are only interested in synoptic fronts (~1000 km), fronts with an along front length ($L$) less than 1000 km are omitted from the analysis (condition C). The frontal line is identified at the maximum of the equivalent potential temperature gradient by applying the following condition:

$$\text{TFP} = 0 \tag{D}$$

The latitude and longitude of where TFP=0 is determined using interpolation. The distance between each adjacent point is calculated and summed across the whole line to give the front length ($L$). The four aforementioned criteria are applied to smoothed $\theta_e$ and horizontal wind fields at 700 hPa in ERA5. The fields are smoothed 30 times using a simple smoothing function whereby the nearest four neighbours of a grid point are averaged and the process is repeated 30 times in order to remove any local-scale features. The same temperature gradient threshold and height level are used as in Schemm et al. (2016), who focused on cold fronts and hail in Switzerland. The 700 hPa pressure level is chosen to avoid inference with orography, which is particularly an issue in central Europe (Jenkner et al., 2010, their section 4.4). Furthermore, the 700 hPa level is further from the turbulent boundary layer. We recognise the 700 hPa level is not the typical height level used by forecasters to identify fronts, but due to the complexities of automatic front detection it is important to derive smooth frontal lines for the cell-front distance calculations and limit the number of erroneously detected fronts. Equivalent potential temperature is selected over potential temperature since it is a function of both temperature and moisture thus gives the strongest frontal signal. One disadvantage to $\theta_e$ is that temperature and humidity can vary independently thus localised humidity gradients could be detected (as discussed in Thomas and Schultz, 2019). However, smaller-scale fronts like sea-breeze, land-breeze, and gust fronts will be filtered out due to the strict 1000 km length threshold and smoothing filter. Therefore, further filtering of local and synoptic fronts, as carried out by Jenkner et al. (2010) and Rüdisühli et al. (2020) for convection-permitting datasets, is not required. The velocity threshold incorporates near-stationary to fast moving cold fronts.

The thresholds chosen (conditions A, B and C) are stricter than some previous studies. Lower thresholds would increase the number of erroneously detected fronts while higher thresholds generally reduce the number of fronts in the dataset but limit the dataset to synoptic-scale cold fronts. We value a dataset with a lower front count and higher percentage of correctly detected fronts. Cold fronts are detected in a subsection of the European domain ([40N–70N, 20W–20E], see Figure 1).

### 2.1.2 Validation

Validation was carried out by manually viewing one year of data and several other case studies to verify that frontal features resembled synoptic-scale fronts (see examples in Figure 2). Figure 1 shows a spatial map with the number of cold-frontal days per warm season between 2007–2016 for April–September to verify plausible spatial distributions. The highest cold-frontal day frequency is over the southern half of the United Kingdom, northern France, the Low Countries and north-western Germany totalling between 20–23 front days per warm season. The minimum is found in the south-east of the domain. However, there is a sample bias for grid points at the edge of the domain since fronts extending beyond the domain may not always meet the 1000 km length criterion. Since we are focusing our analysis on Germany and the surrounding area this does not bias our results. Fronts typically decay as they experience friction from the surface over land. The cold-frontal day maximum is located close to the edge of continental Europe which could be explained by slowly propagating fronts being detected across several days. A boundary is also evident around the Alps which is likely linked to cold fronts becoming distorted and weakening when interacting with orography (Schumann, 1987). This boundary can also be identified in Kunz et al. (2020)'s cold front climatology between 2005–2014 (their Figure 3).

## 2.2   KONRAD Convective Cell Detection and Tracking Algorithm

KONRAD (KONvektionsentwicklung in RADarprodukten; convection evolution in radar products) is a convective cell detection and tracking algorithm (Wapler and James, 2014) originally applied to 2D radar data in the German radar domain (Figure 2). At the start of this study, KONRAD3D (Werner, 2017) was in the final stages of development and was not available over a long time-series so the original KONRAD based on 2D radar data was used. KONRAD is run operationally by the German Weather Service (DWD) with a spatial and time resolution of 1 km and 5 minutes, respectively. A convective cell is defined as one with 15 pixels or more exceeding 46 dBZ. Since the resolution of KONRAD is 1 km, 1 pixel ~ 1 km$^2$. The reflectivity is based on a 0.5 degree radar scan thus the height relative to the ground varies with distance from the radar and due to orography. Where two radar scans overlap the highest dBZ value is used. KONRAD also provides a parameter which indicates the likelihood of hail (the hail flag) ranging from 0 to 2. The parameter is based on cell intensity and area thresholds. A hail flag value of 1 is assigned if at least one pixel in the cell exceeds 55 dBZ. A hail flag value of 2 is assigned if 12 or more pixels exceed 55 dBZ or if at least one pixel exceeds 60 dBZ. Otherwise a hail flag of 0 is assigned. Additional information in relation to the detected convective cells has been made available such as the number of lightning strikes in proximity to the cell centre and mesocyclone characteristics. The lightning strike data are based on the time of arrival method from the European VLF/LF LIghtning detection NETwork (LINET, Betz et al., 2009). A mesocyclone detection algorithm is run operationally by the DWD using doppler radar data (Hengstebeck et al., 2018). The severity of mesocyclones is given a ranking from 1 to 5. We include mesocyclones of ranking 3 or higher in order to include a sample with significant size and mesocyclone strength. To be classified as a mesocyclone with severity level 3 the following criteria must be met: 5 km equivalent diameter, 3 km depth, $10^{-3}$ s$^{-1}$ azimuthal shear or 15 ms$^{-1}$ rotational velocity. A detailed description of the algorithm and practical examples can be found in Hengstebeck et al. (2018). The convective cell, hail flag, lightning and mesocyclone definitions are summarised in Table 1. The number of pixels exceeding 55 dBZ, lightning and mesocyclone definitions will be utilised in section 3.5. However, for all other sections the convective cell definition is based on 15 pixels or more exceeding 46 dBZ. This definition includes cells of moderate to severe intensity. For mid-latitude convective rainfall the Z-R relationship $Z = 300R^{1.4}$ is commonly used (Nelson et al., 2010). A reflectivity of 46 dBZ therefore corresponds to an estimated rain rate of 33 mm hr$^{-1}$. Excluding section 3.5, we consider cells at their first detection time only to avoid repeat counting at additional timesteps in the climatology.

The KONRAD dataset has the following limitations:

- The lifetime of a cell in KONRAD does not reflect the entire lifetime of the cell, rather the time the cell exceeds the reflectivity and area thresholds. The first and last detection time refer to when the cell first and last exceeds the reflectivity and area thresholds.

- Cell splitting and merging can lead to an unrealistic lifetime in some cases (see section 2.1 of Wapler, 2021).

- During a mesoscale convective system (MCS) new cells may be detected at subsequent timesteps and assigned the first detection label due to cell recycling.

- Convective lines are not trivial to identify since a large cell area could indicate a large circular cell or a long and narrow convective line.

## 2.3 Cell and front detection examples

Combining the methods introduced in Section 2.1 and 2.2, four cases are shown in Figure 2 for single timesteps. The cold-frontal line (blue contour) is the result of applying the conditions (A, B and C). Convective cells are considered to be associated with a front in the hour following the front detection. For example, for a front detection at 13 UTC, cells that were first detected between 13:00–13:59 UTC are included. The distance between the convective cell and the 700 hPa frontal line (hereafter cell-front distance) is the distance between the convective cell centre and the nearest grid point on the 700 hPa frontal line contour (i.e., the shortest distance between the frontal line and the convective cell on the surface of an ellipsoidal model of the earth; Karney, 2013). A cell is assigned a positive distance if the average $\theta_e$ surrounding the cell location (4 grid points) is equal to or greater than the frontal line $\theta_e$, thus indicating the cell is on the warm-side of the 700 hPa front. Otherwise, the cell is assigned a negative distance indicating the cell is on the cold-side of the 700 hPa front. Timesteps containing two or more cold-frontal lines in the domain [40N–60N, 5W–20E] were omitted from cell-front distance calculations due to uncertainty deriving the cell-front distance. This is a subdomain of the domain in which fronts were detected (see grey domain in Figure 1). This resulted in 3,506 (18.8%) of the total 18,613 frontal timesteps being omitted. We thus focus on cases where a single large-scale cold front (condition C) is present in the vicinity of Germany that meets the gradient and velocity thresholds (conditions A and B). Omitting timesteps with two or more fronts meeting the criteria does not omit cases with a primary large-scale cold front and a secondary smaller scale front, weaker front or pre-surface-frontal convergence lines. We also note that we do not explicitly consider ana and kata cold fronts separately. During a kata cold front an upper-level humidity front is observed overrunning the surface cold front (Browning and Monk, 1982).

In the examples (Figure 2), situations are shown with cells on the warm-side of the 700 hPa front (hereafter pre-700hPa-frontal), in close proximity to the 700 hPa front (near-700hPa-frontal) and also behind the 700 hPa front (post-700hPa-frontal). In some cases they occur simultaneously in the KONRAD domain (black contour). A NE-SW oriented front is located over Germany in Figure 2a with pre-700hPa-frontal convective cells. A line of cells can be found between 250–400 km ahead of the 700 hPa front. The second case (Figure 2b) shows a cold front that has progressed across Germany and is located in proximity to the Alps. Cells are located post-700hPa-frontal 350–750 km behind the 700 hPa front. Like Figure 2a, Figure 2c also shows a pre-700hPa-frontal case but with a front oriented more towards the N-S direction and with more curvature. A larger number of cells are associated with this case with cells in a larger range of distances between 100–750 km ahead of the 700 hPa front. The final example (Figure 2d) shows a front oriented NE-SW through the middle of the radar domain with both pre-700hPa-frontal and post-700hPa-frontal cells.

## 3 Results

### 3.1 Climatological cold-frontal environments

To understand the importance of different processes and the typical environment across the front, we plot climatological means of four possible drivers of convection depending on the distance from the front: convergence of the wind field at different pressure levels, CAPE, surface dewpoints and the total sky direct solar radiation at the surface (hereafter solar heating; Figure 3). The means are calculated using all instances that an ERA5 grid point has a certain front distance and is not weighted by individual timesteps. The mean of the convergence is shown by the colorbar, whereas the means of CAPE, solar heating and surface dewpoints are normalised between one and zero and shown as black lines. The variable means are normalised to compare the typical magnitude of each variable at different distances from the front. All variables are obtained from ERA5 reanalysis data; the same dataset in which fronts were detected. The means were calculated in the domain of Germany only.

#### 3.1.1 Convergence of the wind field

Convergence and divergence of the wind field is shown by the positive values (red) and negative values (blue), respectively (Figure 3). The strongest climatological near-surface convergence is around 300 km ahead of the 700 hPa front. The result is consistent with understanding of typical cold frontal slopes since the 700 hPa level is typically found ~3 km above the surface and cold fronts have a typical slope of ~1:100 (Bott, 2016b). This indicates that the climatological location of the surface convergence zone, which is linked to the surface front, is 300 km ahead of the 700 hPa front. The surface convergence decreases ahead of the maximum but convergence is present up to 750 km ahead of the 700 hPa front. Near-surface convergence of the wind field is generally linked to lift; a prerequisite for convective initiation (Doswell et al., 1996). However, 200 km behind the maximum (100 km ahead of the 700 hPa front) there is divergence near the surface (blue shading in Figure 3). The strongest climatological divergence is found around 850 hPa under the 700 hPa front location. The implication is that if the cold front location is observed at the 700 hPa level on a synoptic chart the strongest divergence of the wind field can be expected at this location in space. Near-surface divergence of the wind field is generally linked to descending motion, which could inhibit convective initiation.

#### 3.1.2 Convective Available Potential Energy

CAPE is linked to atmospheric instability; a prerequisite for deep moist convection (Doswell et al., 1996). It is also a very commonly used parameter in severe convective storm forecasting. CAPE is climatologically maximum 550–600 km ahead of the 700 hPa front (shown by the dashed line in Figure 3). The highest CAPE available in the environment is therefore typically ahead of the maximum near-surface convergence. CAPE is minimum at the 700 hPa frontal line and slightly increases further behind the 700 hPa front.

### 3.1.3 2-metre dewpoint temperature

Surface dewpoints are linked to the moisture availability at the surface. Figure 3 shows that surface dewpoints are maximum 350–400 km ahead of the 700 hPa front, which is marginally ahead of the maximum climatological near-surface convergence. Further ahead of the 700 hPa front, surface dewpoints decrease but not as quickly as the decrease towards the 700 hPa front. The surface dewpoints reach a minimum after the passage of the 700 hPa front.

### 3.1.4 Total sky direct solar radiation at the surface

Solar heating is linked to increased surface temperatures, which contributes towards atmospheric instability and lifting of air parcels near the surface. Solar heating is minimum 100–150 km ahead of the 700 hPa front (solid line with circular markers in Figure 3), which indicates increased cloudiness. The maximum in large-scale precipitation in ERA5 was also found at this distance relative to the front (not shown). Further ahead of the 700 hPa front solar radiation gradually increases reaching a maximum 700–750 ahead of the 700 hPa front. The increase in the other direction towards the post-700hPa-frontal environment is sharper, which can be explained by cloudiness of the warm-sector whereas the post-700hPa-frontal environment remains largely cloud-free.

### 3.1.5 Surface front relative to the 700 hPa front

Since near-surface convergence is relevant for lifting parcels to their level of free convection (one of the ingredients for deep moist convection; Doswell et al., 1996), the strongest climatological near-surface convergence (Figure 3) is used as a reference point for the surface front location relative to the 700 hPa front. Furthermore, the surface front would be expected to be 300 km ahead of the 700 hPa front since the 700 hPa level is typically found ~3 km above the surface and cold fronts have a typical slope of ~1:100. While the slope of the front and corresponding surface front location relative to the 700 hPa front are likely to vary slightly per case study, we proceed assuming the climatological location of the surface front (hereafter surface front) is 300 km ahead of the 700 hPa front for this study.

## 3.2 Convective cell count and days

During the period 2007–2016 for the months April–September, 258,218 cells were detected in KONRAD (Section 2.2). On cold-frontal cell days, 163,255 cells (63%) were detected (see Table 2). A cold-frontal cell day is defined as one where at least one convective cell was detected within 750 km of the 700 hPa frontal line. The remaining 94,963 (37%) cells were detected on non-cold-frontal days. A cell was associated with a cold front on 614 days, thus approximately a third of warm-season days are associated with cold-frontal convection in Germany. This does not imply a new frontal system every 3 days as many cases are associated with slowly propagating fronts across several days. While a slightly larger percentage of all cell days were non-cold-frontal cell days (56%), over twice as many cells developed on average on cold-frontal cell days (266 cells per day) compared to non-cold-frontal cell days (123 cells per day) (Figure 4). Considering all 1389 days convective cells were detected in KONRAD, the 95th percentile of the cells per day is 725 accounting for 80 days; 80% of these days occurred on cold-frontal

cell days. This highlights that extreme events (in terms of cell count per day) are significantly more likely on cold-frontal cell days. The differences in the cell per day count could reflect convection being more widespread across Germany and/or more cells associated with each cell cluster. During an MCS for example a larger amount of cells are typically detected in KONRAD as a continuous line exceeding 46 dBZ may not always be present so several cells are detected within the MCS. Secondly, due to cell recycling in an MCS, KONRAD will likely detect new cells at subsequent timesteps and assign them the first detection label. Mesoscale convective systems could therefore also explain the large differences in the cell count per day between cold-frontal cell days and non-cold-frontal cell days. These results are consistent with findings from Wapler and James (2014) who found the synoptic patterns with the highest number of convective cells were Cyclonic Southerly, Anticyclonic South-Easterly, Cyclonic South- Westerly and Trough over Western Europe. Such synoptic patterns are likely to indicate the presence of a cold front excluding Anticyclonic South-Easterly. The cell statistics are summarised in Table 2.

Figure 5a shows the number of convective cells (blue) depending on the cell-front distance. Positive distances indicate pre-700hPa-frontal cells and negative distances post-700hPa-frontal cells. A total of 65,567 cells were found in proximity (±750 km) to the 700 hPa frontal line between 2007–2016 (April–September). The 65,567 cells are a subset of the 163,255 cells on cold-frontal cell days, the remaining cells occurred outside of the 750 km range, on timesteps with two or more fronts or on timesteps with no cold front detected. The most common region for cells is pre-700hPa-frontal accounting for 84% of all cells. The highest cell frequency is 350–400 km ahead of the 700 hPa front. This corresponds to the location marginally ahead of the mean surface front location (section 3.1.5) and corresponds to the where surface dewpoints are climatologically highest (Figure 3). Further ahead of the front the cell frequency decreases but remains significantly higher than post-700hPa-frontal cells. Furthermore, a slight increase in the cell frequency is observed around 650–700 km ahead of the 700 hPa front which could be linked to pre-surface-frontal convergence lines (Dahl and Fischer, 2016). Behind the surface front the cell frequency steeply declines towards the 700 hPa front. The 700 hPa frontal line corresponds to the minimum cell frequency. This also justifies our use of the 700 hPa level since it marks a clear shift in the probability of convective cells. This minimum could be linked to the descent observed behind a cold front (Davies and Wernli, 2015), as also shown by the maximum climatological divergence in the lower levels (Figure 3). Furthermore, the lowest climatological CAPE is at the 700 hPa front location. Synoptic-scale fronts are typically analysed by forecasters at the surface or 850 hPa, thus forecasters should not expect the lowest convective activity to be after the front has passed at 850 hPa or the surface, rather after the frontal passage at 700 hPa. Post-700hPa-frontal cells account for 16% of all cells. Similar to the pre-700hPa-frontal environment, there is a relatively normalised distribution around the maximum frequency 350–400 km behind the front. Assuming post-700hPa-frontal convection is primarily driven by solar heating, a certain period is likely required after the front passes before convective initiation is favourable. Figure 3 highlights that solar heating increases sharply during and after the passage of the 700 hPa front and CAPE also increases slightly. Lightning strike frequency exhibits a similar distribution around the front in Germany (Figure A1) and in a sub-European domain (Figure A2). This similarity indicates that the distribution around the front observed in Figure 5a is not confined to Germany nor to the KONRAD convective cell definitions.

The number of convective cell days is shown in Figure 5b. The number of convective cell days refers to the number of days a cell was found between each 50 km interval shown in Figure 5. A similar distribution around the front is observed as in

Figure 5a but with a slight shift in the maximum further from the 700 hPa front. This is true both post-700hPa-frontal and pre-700hPa-frontal with the new maximum found -500 to -450 km and 400 to 450 km from the 700 hPa front, respectively. While approximately 5 times as many cells were found around the pre-700hPa-frontal maximum (~5000 cells) compared to the post-700hPa-frontal maximum (~1000 cells), the number of cell days differs by a factor of 2.5. Therefore, pre-700hPa-frontal cases are associated with a greater number of cells than post-700hPa-frontal cases which explains the significant difference in the magnitude of the peak when looking at the total number of cells.

Since cold fronts typically weaken as they move over continental Europe there is possibility of a sample bias with more pre-700hPa-frontal environments being sampled than post-700hPa-frontal environments. This bias was assessed by shuffling each cold-frontal timestep in the analysis period to a random timestep and then deriving the cell-front distances again. This process was repeated 100 times and a new cell count around the front was produced using the random sample (see Figure B1). The sample bias can be seen for pre-700hPa-frontal environments, however when dividing the actual cell count by the random sample cell count we see no significant differences in the distribution around the 700 hPa front compared to Figure 5a. Figure B1 also highlights that between 100–1000 km ahead of the 700 hPa front the cell count is higher than the random sample cell count.

## 3.3 Diurnal Cycle

Convection is known to generally exhibit a diurnal cycle with a maximum in the afternoon hours and a minimum in the early morning hours (Ban et al., 2014). Figure 6 shows the cell frequency as a function of the cell-front distance and hour of the day. As noted in Figure 5, the maximum cell count is found pre-700hPa-frontal with the minimum at the 700 hPa front. Figure 6 highlights that most post-700hPa-frontal cells develop during the daytime hours (especially between 10–18 UTC) with low cell frequency outside of this period. In contrast, pre-700hPa-frontal cells do develop in the night hours and the cell maximum is shifted later in the day. The exception to this is further from the front in the region 750–1000 km where cells have a stronger diurnal cycle. We speculate that the lifting associated with the front is weaker this far from the front thus cells are largely diurnally driven. This result also justifies our distance criterion at which we consider a convective cell to be associated with a front, that is, within 750 km of the 700 hPa frontal line.

Figure 7 shows the same data as Figure 6 but with an average taken across the pre-700hPa-frontal (0 to 750 km) and post-700hPa-frontal (-750 to 0 km) environments. The diurnal cycle of non-cold-frontal cells (green) is shown for reference. A strong diurnal cycle for post-700hPa-frontal cells (blue line) is evident with a maximum between 15–16 UTC. The distribution for non-cold-frontal cells is similar with a maximum also between 15–16 UTC. The pre-700hPa-frontal maximum is found between 16–17 UTC, a later peak than in the non-cold-frontal and post-700hPa-frontal categories. The weakened cycle is evident by the larger percentage of cells occurring during the night-time hours. Between 21–06 UTC, 31% of pre-700hPa-frontal cells developed compared to only 9% and 13% of post-700hPa-frontal and non-cold-frontal cells, respectively. A similar weakening of the diurnal cycle was observed for mesocyclones compared to lightning in Germany (Wapler et al., 2016). This indicates that supercells, where a deep and persistent mesocyclone is present, may also be linked to the observed weakened diurnal cycle. Mesocyclone frequency depending on the cell-front distance will be investigated in section 3.5. Furthermore, Morel and

Senesi (2002) found that MCSs dissipate most commonly at 21 Local Solar Time (LST) on average in Europe. Surowiecki and Taszarek (2020) found a similar result showing the majority of MCSs dissipate around 19–20UTC in Poland, though squall line/bow echo MCSs most frequently dissipate at midnight. MCSs typically live longer than individual cells, and new cells may be triggered throughout their lifecycle, e.g., through interaction with the cold pool which is less dependant on solar heating. We speculate therefore that the observed pre-700hPa-frontal weakened diurnal cycle could be linked to mesoscale convective systems. We also investigated the monthly cycle of convective cells depending on the cell-front distance but no significant differences were found with the peak season being mid-July for cold-frontal cells. The peak season for non-cold-frontal cells was early July.

## 3.4 Spatial cell climatology

The frequency of convective events varies in different parts of Europe (Groenemeijer et al., 2017; Taszarek et al., 2019 and 2020a). Schemm et al. (2016) also showed the fraction of hail events associated with a synoptic-scale cold front varied spatially across Switzerland and the vicinity. In this section we produce spatial maps of convective cell days per warm season in the KONRAD radar domain (grey contour in Figure 8). As in Figure 4, a cell day is defined as one where at least one convective cell was detected, in this case within a 1 x 1 degree grid box.

### 3.4.1 Convective cell days

Distinct spatial distributions were found for pre-700hPa-frontal and post-700hPa-frontal cells (Figure 8). Post-700hPa-frontal cells (Figure 8a) are most frequent in north-west Germany developing between 5–9 days per warm season. The highest frequency is near the coastline, which could be linked to sea-breezes, higher moisture availability and increased lift with cooler air being advected over a warmer surface. Moving south and east the number of cells per warm season reduces to 1–3. Pre-700hPa-frontal cells (Figure 8b) are most frequent in the southern half of Germany, particularly in the far south near the borders with France, Switzerland and Austria. Cells develop between 8–16 days per warm season in the majority of the domain in the pre-700hPa-frontal environment. The exceptions are grid boxes outside of Germany which are likely under-sampled as they are located further from radars. The combined pre-700hPa-frontal and post-700hPa-frontal convective cell spatial distribution shows one maximum in southern Germany and another in north-west Germany (Figure 8c). North-east Germany receives the lowest number of cold-frontal cell days per warm season. The non-cold-frontal (Figure 8d) and all cells spatial distributions (Figure 8e) are very similar to pre-700hPa-frontal cells (Figure 8b) with a maximum cell frequency in southern Germany. The high cell frequency in southern Germany is largely linked to the orography in this region which can modify the mesoscale circulation and environment (see elevation contours on Figure 8f). Several studies have shown that thunderstorms are more frequent near mountainous regions (e.g., Piper and Kunz, 2017; Taszarek et al., 2019).

### 3.4.2 Cold-frontal cell day fraction

Figure 8f shows the percentage of all convective cell days associated with cold fronts for 1 x 1 degree grid boxes. Around 35–40% of convective cell days were associated with cold fronts for much of north-western Germany. This could be linked to the high frequency of cold fronts in this region (Figure 1). The percentage decreases moving further south and east. In south-
eastern and eastern Germany, western Czechia and western Poland only 20–30% of cell days were cold-frontal. These regions are mostly in proximity to elevated terrain (see contours on Figure 8f). Kunz et al. (2020) found a similar result showing that over complex terrains, the proportion of frontal severe convective storms is around 10–15%, while over flat terrain 50% require a frontal trigger.

### 3.4.3 Cold front types

The orientation and geographical location of cold fronts could potentially be used as an indicator for the likelihood of cells. K-means clustering, an unsupervised machine learning algorithm (section 14.3.1 in Wilks, 2006), is used to cluster frontal timesteps into clusters such that each cluster contains fronts with similar characteristics (i.e., spatially located and orientated in a similar way). A cropped domain ([40N–60N,0–20E]) of the original front detection domain (Figure 1) was used. A total of 4,230 frontal timesteps with convective cells were available to be clustered. Since the input into the algorithm is binary in
our case (i.e., 1 for a cold-frontal grid point and 0 for a non-cold-frontal grid point) no normalisation is required.

K-means clustering requires the cluster number to be predefined. The elbow method and silhouette score were first tested between cluster numbers 2 and 50 (see Figure C1) to determine the optimal cluster number. However, no clear elbow could be identified, which is often the case for many real world problems. The highest silhouette score was for 2 clusters which consisted of a cluster with fronts close to the Alps and another cluster with fronts to the north-west of Germany. However, high
variance was found within both clusters and no clear orientation of the front could be identified indicating several different fronts were contained in each cluster. Since the elbow method and silhouette score proved ineffective, plots of cluster numbers 10 through 50 were produced to find the optimal cluster number in terms of not too many clusters resulting in large similarities between clusters (and the front type) but not too little resulting in fronts with different characteristics being grouped in the same cluster. Thirty was selected subjectively as the number of clusters based on viewing the aforementioned plots. Six clusters
(1,133 timesteps) were removed due to a lack of continuity in the clusters. We draw frontal contours (Figure 9) where 50% of timesteps in the respective cluster had the front located at that grid point. Plots for cluster numbers 15, 35 and 50 are shown in the Appendix (Figures C2, C3 and C4).

The resulting 24 front clusters are shown in Figure 9 sorted by the overall largest number of convective cells per timestep. The colorbar represents the number of cells per timestep per 1 x 1 degree grid box. Cluster 1 has the highest cell frequency, which
is associated with a NE-SW oriented front extending from Scandinavia to France. The highest cell frequency is in the western half of Germany ahead of the 700 hPa front, consistent with our results shown in Figure 5. The convective cell minimum around the 700 hPa frontal location can also be located in several clusters (e.g., Clusters 11, 15 and 20). Cluster 2 and 3 show similar front types to Cluster 1 but are positioned slightly further west (Cluster 2) or east (Cluster 3). Post-700hPa-frontal

cells are mostly attributed to clusters 13, 20, 22, 23 and 24 when the front has progressed further inland across Germany. A period of solar heating is likely required after the front passes before convective initiation is favourable. Clusters 22, 23 and 24 also highlight how cold fronts become deformed on their interaction with the Alps (Schumann, 1987). Cluster 17 is worthy of mentioning due to the high cell frequency per timestep near the German-Czech border. Piper and Kunz (2017) found increased thunderstorm activity in this region based on lightning data between 2001–2014. These plots provide useful insight for forecasters highlighting where in respect to the front cells are most likely for certain front types and their corresponding geographical locations.

## 3.5 The nature of convective cells

As mentioned in Section 2.2, KONRAD defines a convective cell based on reflectivity and area thresholds. Namely, a convective cell is defined as a cell with 15 pixels (~15 km$^2$) or more exceeding 46 dBZ (see Table 1). Furthermore, a hail flag value of 1 is assigned if at least one pixel in the cell exceeds 55 dBZ. A hail flag value of 2 is assigned if 12 or more pixels exceed 55 dBZ or if at least one pixel exceeds 60 dBZ. Using the same definitions of cold-frontal and non-cold-frontal cell days as in Figure 4, Figure 10 highlights that mostly all cell days have at least one cell with the hail flag value 1; 96% of cold-frontal cell days and 87% of non-cold-frontal cell days. However, with increasing number of pixels a discrepancy is found. For example, 69% of cold-frontal cell days have a cell with 12 pixels exceeding 55 dBZ (similar criterion for hail flag 2), opposed to only 46% of non-cold-frontal cell days. The discrepancy becomes larger for increasing number of pixels. Around 30% of cold-frontal cell days have a cell with 50 pixels exceeding 55 dBZ, opposed to only 10% of non-cold-frontal days. These results highlight that given cells initiate in proximity to a cold front, cells with larger intense convective cores are more likely to develop. These larger cores could be linked to an increased number of strong convective lines present on cold-frontal cell days. The 55 dBZ threshold has also been proposed as a reflectivity threshold for hail (Mason, 1971). Several studies have evaluated the threshold (e.g., Hohl et al., 2002; Wapler et al., 2012; Kunz and Kugel, 2015). More recently, Wapler (2017) validated the hail flag against ESWD reports showing that 72% of 2 cm or larger hail reports and 84% of 5 cm or larger hail reports were labelled hail flag 2. Wapler (2017) also found that the synoptic pattern most commonly associated with hail in Germany is a cyclonic south-westerly flow. Such a synoptic pattern is often associated with a cold front and sometimes the advection of an elevated mixed layer originating from North Africa or the Spanish plateau (Lewis and Gray, 2010; Dahl and Fischer, 2016). The steep lapse rates in the midlevels can contribute to atmospheric instability which is one factor relevant for large hail (Púčik et al., 2015; Taszarek et al., 2020b). Schemm et al. (2016) found up to 45% of hail events in north-eastern and southern Switzerland formed in pre-frontal zones. They argued that vertical wind shear and along frontal moisture transport were among the mechanisms favouring hail formation in pre-frontal environments. Additionally, Kunz et al. (2020) showed frontal hailstorms produce larger hail and have a longer track on average. Future studies should utilise hail reports such as those collected in the ESWD and automatic front detection on a pan-European scale to better understand the link between cold fronts and hail in Europe.

Figure 11 shows several characteristics of convective cells depending on the cell-front distance. The average for non-cold-frontal cells (dashed green line) is shown for reference. The longest mean cell lifetime (Figure 11a) of around 20 minutes is found 450–750 km ahead of the 700 hPa front, thus pre-surface-frontal. For the first cell detection the lifetime assigned is 5 minutes and for subsequent timesteps 5 minutes is added. Pre-700hPa-frontal cells further than 150 km from the 700 hPa front have a longer lifetime than the non-cold-frontal 18 minute average. Post-700hPa-frontal and near-700hPa-frontal cells have the shortest lifetime. Due to the convective cell definition of 46 dBZ (15 pixels), the cell lifetime refers to the time a cell exceeds these thresholds rather than the overall lifetime of the cell, thus the actual lifetime of the convection transitioning from the development, mature and decay phase would be longer. Cell splitting and merging can also lead to an unrealistic lifetime in some cases (see section 2.1 of Wapler, 2021).

Wapler and James (2014) showed that convective cells in Germany propagate in different directions and with different speeds depending on the synoptic pattern. The cell speed is important as it can favour the likelihood of certain convective hazards. Slowly propagating convective systems typically lead to an increased chance of flooding. For example, a series of slowly moving thunderstorms under weak flow led to severe flooding in Germany in 2006 (Piper et al., 2016). On the other hand, faster propagating systems are more likely to be associated with severe convective winds. For example, Gatzen et al. (2020) showed that both warm and cold season bow echoes in Germany occurred under strong 500 hPa flow. In cold-frontal environments we find cells propagate faster than non-cold-frontal cells at all locations relative to the front (Figure 11b). The maximum mean cell speed is found 150–250 km ahead of the 700 hPa front. Two minimums are found, one 550–650 km ahead of the 700 hPa front (pre-surface-frontal) and 650–750 km behind the 700 hPa front.

Once convection has initiated, secondary convection is often found in proximity due to interactions between cell outflows and other boundaries (Wulfmeyer et al., 2011). Radar and satellite observations highlight that post-frontal convective cells are typically scattered in nature (Theusner and Hauf, 2004; Weusthoff and Hauf, 2008). Theusner and Hauf (2004) showed that 72% of post-700hPa-frontal cells were single cells based on 39 warm-season days in Germany. We calculate the number of cells present within a 100 km radius at the detection time to provide an estimate of the aforementioned interactions (Figure 11c). We find that pre-surface-frontal cells (350–750 km) have the largest number of cells present in a 100 km radius (around 4 other cells). Cell outflow boundaries interacting with other surface convergence lines such as the surface front and pre-surface-frontal convergence lines could result in increased convergence and the initiation of new cells. The minimum is found near-700hPa-frontal. where only 1 additional cell is present on average. Post-700hPa-frontal cells are also associated with a smaller amount of cells in proximity (1–1.5 cells) compared to pre-700hPa-frontal cells and non-frontal cells (2.25 cells).

Figure 11d indicates the fraction of cells associated with lightning depending on the cell-front distance. A cell is considered to be associated with lightning if 3 or more lightning strokes are detected 20 km or less from the cell centre during the cell's lifetime. We require at least 3 strokes to filter out any erroneous detected strokes, as was carried by Púčik et al. (2015) to confirm presence of a thunderstorm. The lightning fraction maximum is found 550–650 km ahead of the 700 hPa front. Other pre-surface-frontal cells are also commonly associated with lightning (around 80% of cells). The fraction of cells associated with lightning sharply declines with increasing proximity to the 700 hPa cold front. The fraction of cells with lightning increases again in the post-700hPa-frontal environment but the overall fraction with lightning is lower than most pre-700hPa-frontal cells

and the non-frontal fraction (0.73). Observational studies have indicated that low instability in the lower mixed-phase region can discriminate between convection associated with lightning and non-lightning convection (e.g., van den Broeke et al., 2005). Differences in the typical vertical profile could vary across the front explaining the observed differences in lightning probability.

Figure 3 shows that across the front CAPE is climatologically lowest near the 700 hPa front corresponding to the minimum cell lightning fraction and highest 550–600 km ahead of the 700 hPa front corresponding to the highest cell lightning fraction.

Figure 11e is complementary to Figure 10 highlighting it is generally pre-surface-frontal cells that have an increased likelihood of becoming intense compared to non-cold-frontal cells. Like the cell lifetime, cells further than 150 km ahead of the 700 hPa front surpass the non-cold-frontal average. The largest fraction with the hail flag (around 12%) is cells 550–650 km ahead

of the 700 hPa front, around 200 km ahead of the cell frequency maximum (Figure 5), and where CAPE is climatologically highest (Figure 3). As with the lightning fraction, near-700hPa-frontal cells have the lowest fraction associated with the hail flag 2. Following the result that near-700hPa-frontal cells are typically less intense one may suspect the lower lightning fraction is simply related to cell intensity. However, upon considering cells that only meet the minimum cell threshold (15 pixels at 46 dBZ and no pixels at 55 dBZ), the same distribution was found with the lowest lightning fraction near-700hPa-frontal and

the highest pre-surface-frontal. The largest post-700-hPa-frontal hail flag 2 fraction is found 250–450 km behind the 700 hPa front, closely corresponding to the maximum frequency of post-700 hPa frontal cells (Figure 5).

Mesoyclones, where a rotating updraft is present, are known to be linked to severe convective hazards such as hail and tornadoes. For example, Wapler (2017) showed 75% of hail events in Germany were associated with mesocyclones. Based on the mesocyclone criteria introduced in section 2.2 (see Table 1), around 5% and 1.5% of pre-surface-frontal cells and

490 non-cold-frontal cells had mesocyclones, respectively. Near-700hPa-frontal cells are also more likely to be associated with a mesocyclone than non-cold-frontal or post-700hPa-frontal cells. A key result from this section is that while convective cells are most frequent near the surface front (Figure 5a), when they do develop further ahead of the surface front they are as likely or more likely to be associated with lightning, 55 dBZ cores and mesocyclones, especially in comparison to non-cold-frontal cells.

## 495 4 Conclusion

A novel climatology of cold-frontal convective cells in the German radar domain (Figure 2) has been produced using automatic front detection methods (section 2.1) and a convective cell detection and tracking dataset (section 2.2) between 2007–2016 for April–September. Furthermore, the nature of cells such as the intensity, lightning frequency and mesocyclone frequency has been analysed. Previous cold-frontal convection studies in Europe had primarily focused on narrow cold-frontal rainbands

(Gatzen, 2011; Clark, 2013) or on cold-frontal hailstorms (Schemm et al., 2016; Kunz et al., 2020). In this study, we focused on the nature and climatological differences of convective cells depending on their distance from the front as well as making a comparison with non-cold-frontal cells.

The primary findings of the study are outlined below:

– Cold-frontal cell days are associated with around twice as many cells on average compared to non-cold-frontal cell days (Figure 4). Cold-frontal cell days are more likely to be associated with cells with large 55 dBZ cores. Around 30% of cold-frontal convective cell days were associated with at least one cell with a 50 km$^2$ area exceeding 55 dBZ opposed to only 10% of non-cold-frontal cell days (Figure 10).

   – Convective cells are most frequent 350–400 km ahead of the 700 hPa front (Figure 5a), which is marginally ahead of the
typical surface front location (section 3.1.5; Figure 3). Surface dewpoints are highest in the same region (Figure 3).

   – The minimum cell frequency is directly at the 700 hPa front with over 5 times less cells developing compared to the 350–400 km maximum (Figure 5a). The minimum CAPE is found at the 700 hPa front and the strongest divergence in the lower levels (Figure 3).

   – Post-700hPa-frontal cells most commonly develop 350–400 km behind the 700 hPa front (Figure 5a), where solar heating
increases after the passing of the 700 hPa front and CAPE also increases slightly (Figure 3).

   – Post-700hPa-frontal and non-cold-frontal cells exhibit a typical convective diurnal cycle with a maximum between 15–16 UTC. Pre-700hPa-frontal cells exhibit a weakened diurnal cycle with around a third of all cells developing between 21–06 UTC (Figures 6 and 7). The weakened cycle is consistent with observations of mesoscale convective systems and mesocyclones in Europe.

– Post-700hPa-frontal cells are most frequent in north-west Germany, particularly close to the coast (Figure 8a). Pre-700hPa-frontal and non-cold-frontal cells show similar spatial distributions with cells most frequent in the southern half of Germany (Figures 8b and 8d). Cells in south-east and eastern Germany are less associated with cold fronts than other regions (Figure 8f). Cold-frontal cell frequency is highest in Germany when the 700 hPa front is orientated NE-SW to the west or north-west of Germany (Figure 9).

– Pre-700hPa-frontal cells (particularly pre-surface-frontal cells) are longer-living and more often associated with 55 dBZ cores, lightning and mesocyclones than cells in other categories. The opposite is true for post-700hPa-frontal cells compared to non-cold-frontal cells excluding mesocyclones where less significant differences were found between the non-cold-frontal and post-700hPa-frontal fractions. CAPE is climatologically highest ahead of the surface front (Figure 3) providing some indication as to why a higher fraction of pre-surface-frontal cells have 55 dBZ cores, lightning and
mesocyclones (Figure 11). Cells at all distances relative to the front propagate faster on average than non-cold-frontal cells (Figure 11).

These results act as an important reference for future studies on cold-frontal convection. Since our results have shown the climatology and nature of convective cells vary substantially in Germany depending on their cell-front distance and in comparison to non-cold-frontal cells, this research should prompt further studies on cold-frontal convection especially regarding
links to convective hazards. Future studies should also be performed on a pan-European scale to highlight any potential spatial differences in cold-frontal convection climatology.

*Code and data availability.* ERA5 data can be downloaded from the Copernicus servers (ERA5). KONRAD is available for research purposes on request (contact kundenservice@dwd.de). Front detection and plotting code are available on request from the corresponding author.

*Author contributions.* SP and LS wrote the original research proposal. GP carried out the data analysis and wrote all sections of the manuscript. SP, LS and KW provided comments and support during the data analysis and manuscript process. KW provided technical support with the KONRAD cell detection and tracking dataset.

*Competing interests.* The authors declare there are no competing interests

*Acknowledgements.* This research has been funded by Deutsche Forschungsgemeinschaft (DFG) through grant CRC 1114 'Scaling Cascades in Complex Systems, Project Number 235221301', Project C06 "Multi-scale structure of atmospheric vortices". We thank three anonymous
reviewers whose comments greatly improved the quality of the manuscript. We also thank Stefan Rüdisühli for useful discussions and insight regarding automatic front detection methods as well as Johannes Dahl who helped interpret certain results.

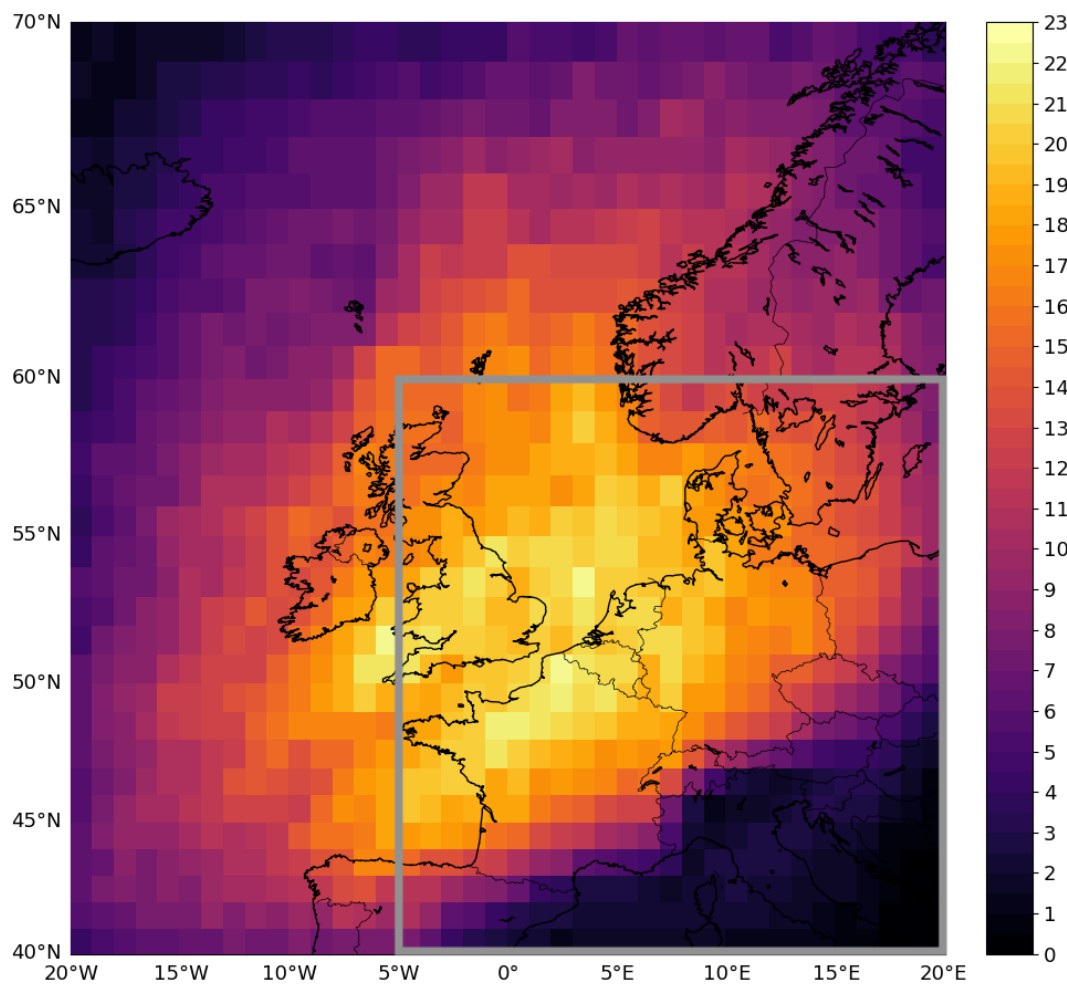

**Figure 1.** Number of cold-frontal days per warm-season between 2007–2016 for 1 x 1 degree grid boxes in the domain [40N-70N, 20W–20E]. The domain corresponds to the domain in which cold fronts were detected. If two or more fronts were detected in the grey domain ([40N-60N, 5W-20E]) at a given timestep the timestep was omitted from cell-distance calculations (see discussion is section 2.3). The marked grey domain also represents the domain in which lightning data were used for the data presented in Figure A2.

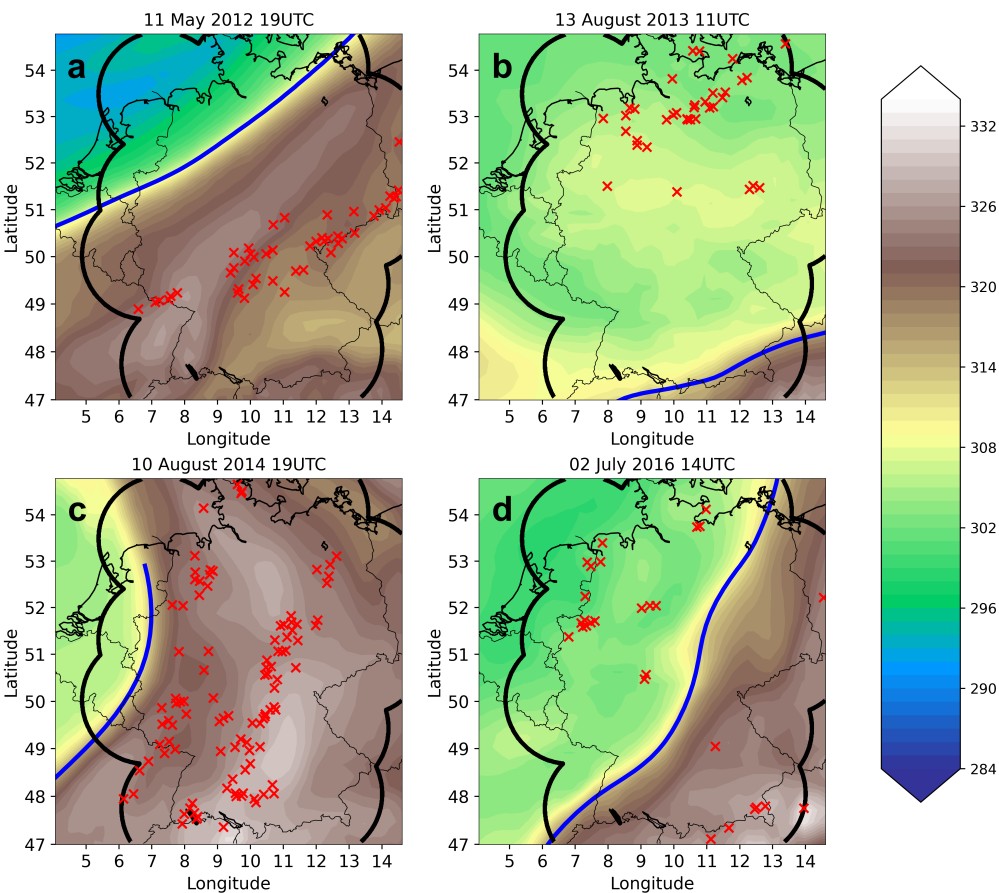

**Figure 2.** 700 hPa cold-frontal line (blue), convective cells (red crosses) and 700 hPa equivalent potential temperature in kelvin (shaded) for four cases at single timesteps. The black contour shows the KONRAD convective cell domain.

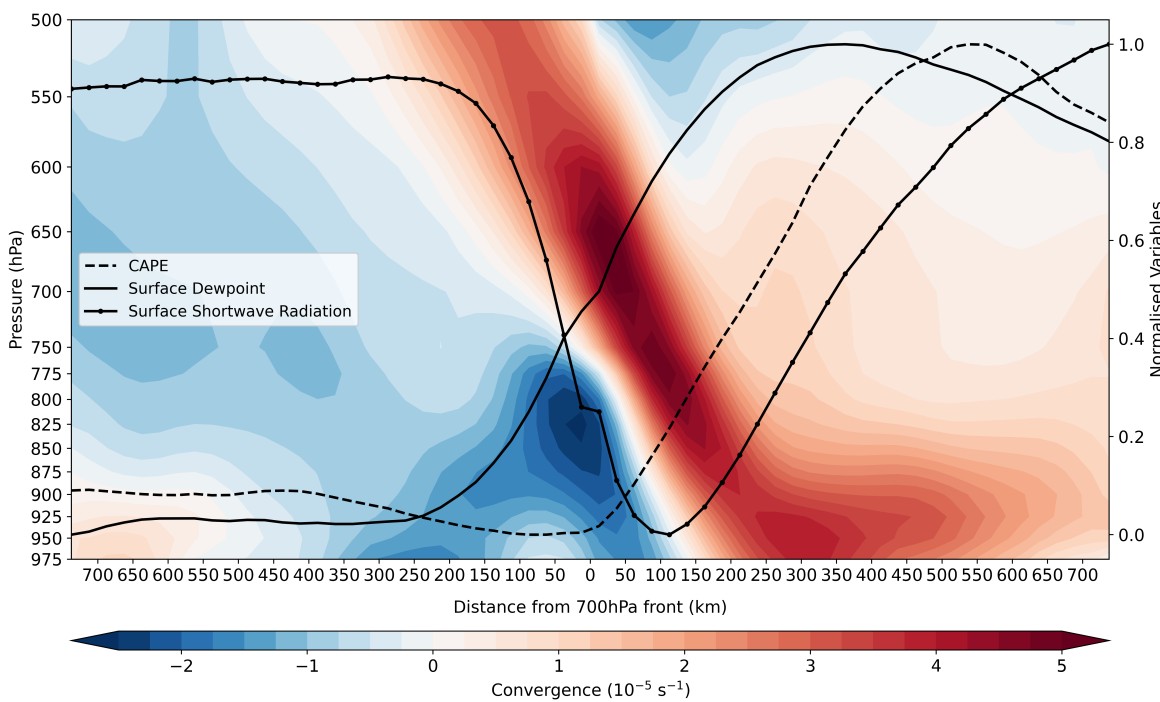

**Figure 3.** Climatological convergence at pressure levels (shaded), CAPE (dashed line), surface dewpoints (solid line), surface shortwave radiation (solid line with circular markers) in ERA5 depending on the distance from the 700 hPa front. Excluding convergence, the climatological means are normalised between 0 and 1. The CAPE in ERA5 is calculated based on the most unstable parcel below 350 hPa.

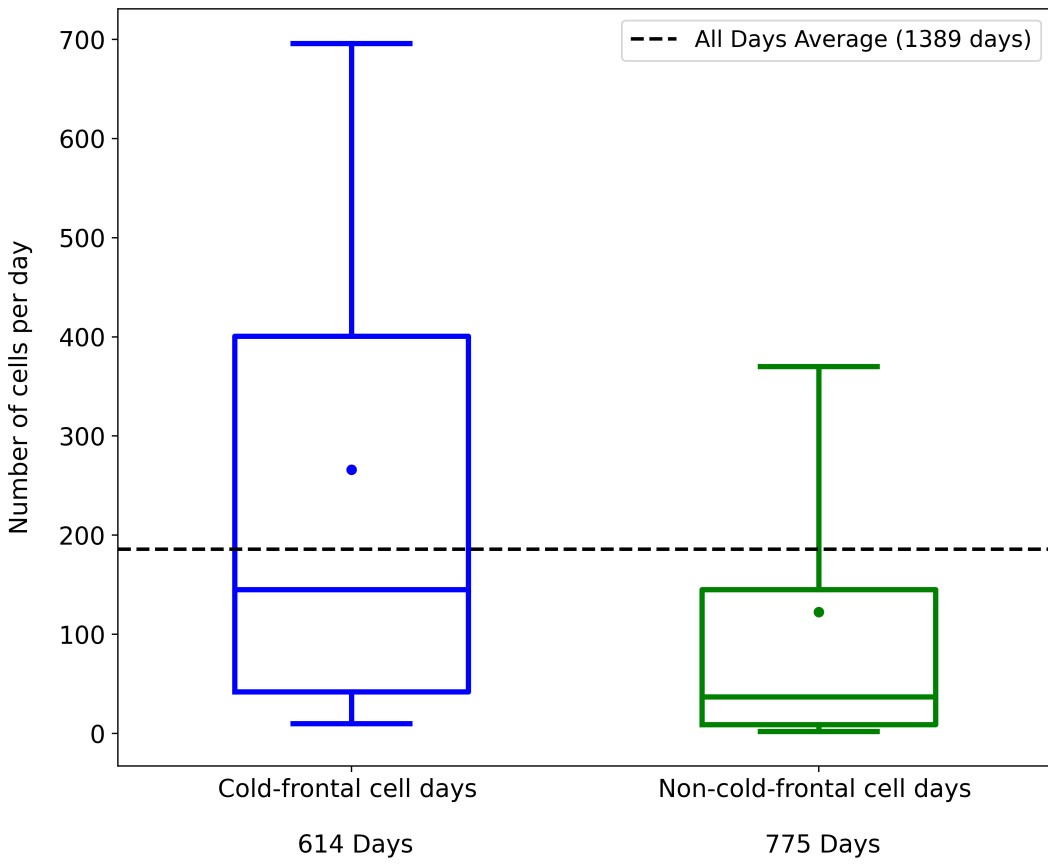

**Figure 4.** Number of cells per day boxplots for cold-frontal cell days and non-cold-frontal cell days. A cold-frontal cell day is defined as one where at least one convective cell was detected within 750 km of the 700 hPa frontal line, otherwise the day is classified as a non-cold-frontal cell day. The median is represented as a horizontal line, dots represent the mean, boxes represent the 25th–75th percentile values, and whiskers represent the 10th and 90th percentile values. The all days mean (1389 days) is shown by the dashed black line.

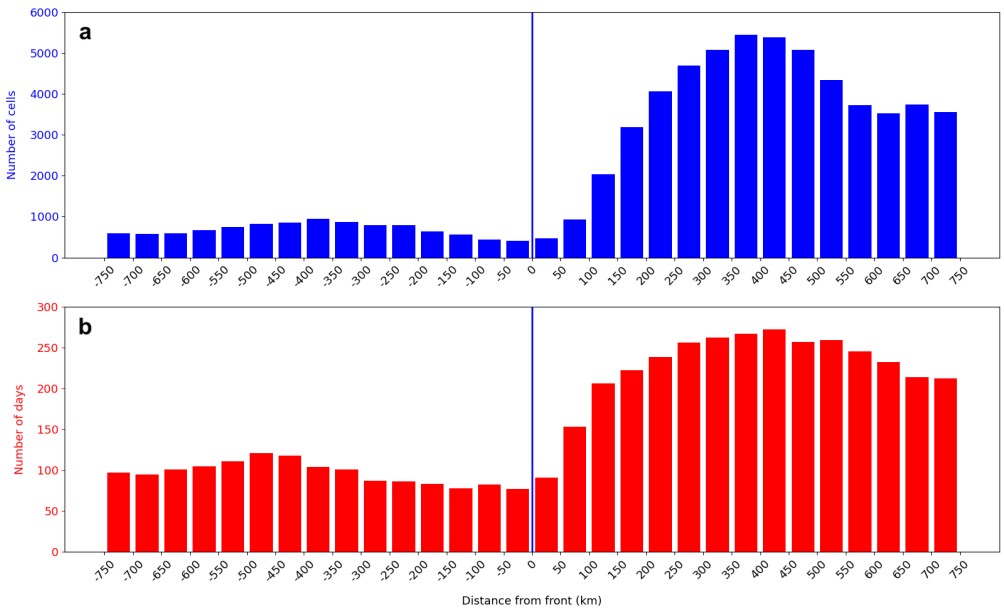

**Figure 5.** Number of convective cells (a) and number of convective cell days (b) depending on the cell-front distance. Positive and negative distances represent the pre-700hPa-frontal and post-700hPa-frontal environments, respectively.

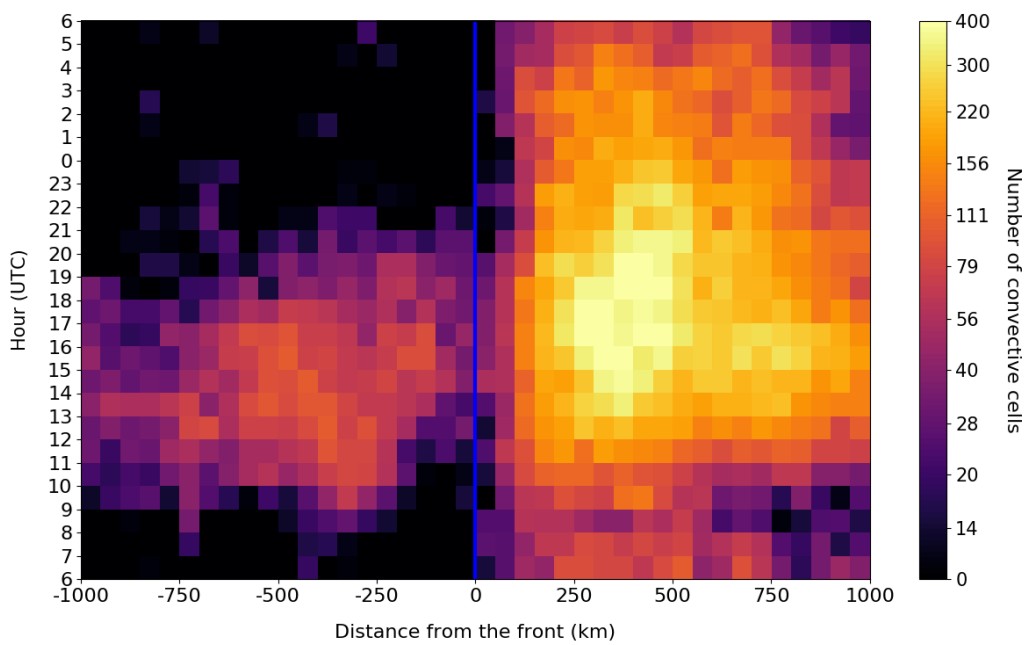

**Figure 6.** Number of convective cells (non-linear colorbar) depending on the cell-front distance (horizontal axis) and hour of the day (vertical axis). Negative distances indicate post-700hPa-frontal cells while positive distances indicate pre-700hPa-frontal cells.

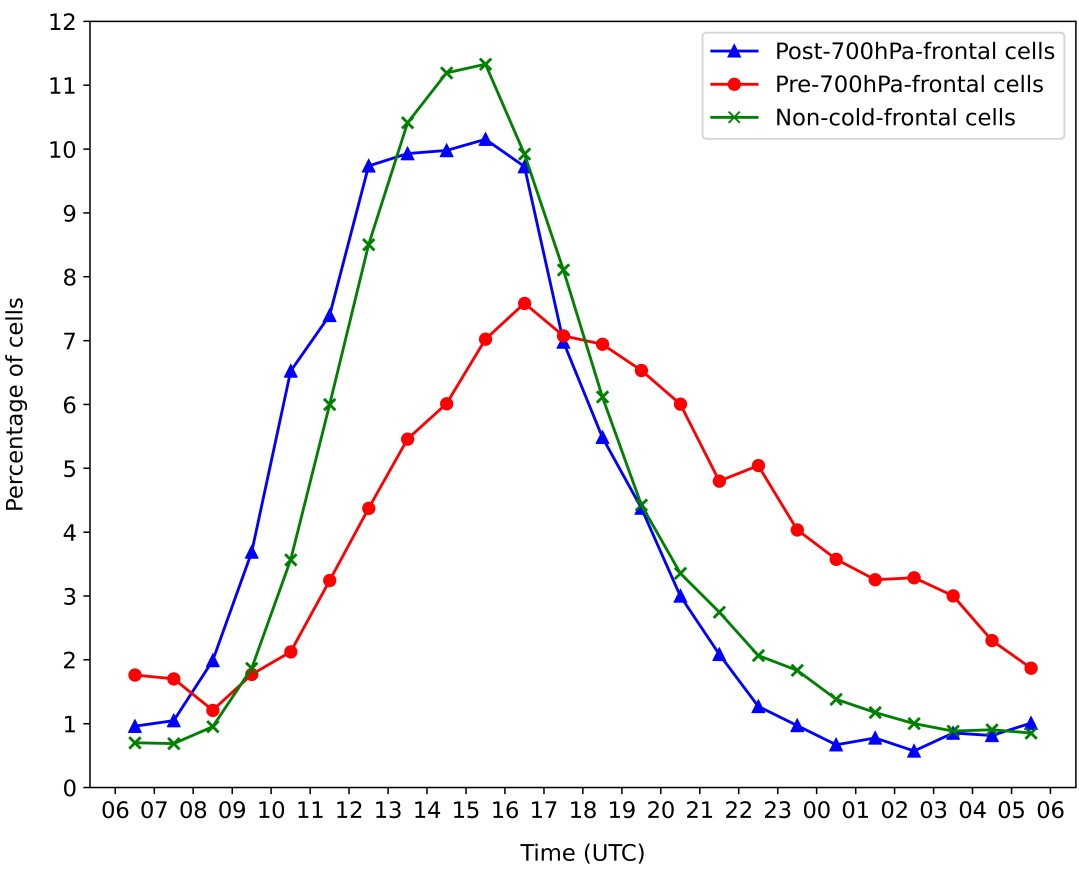

**Figure 7.** Diurnal cycle of convective cells for post-700hPa-frontal (blue), pre-700hPa-frontal (red) and non-cold-frontal cells (green).

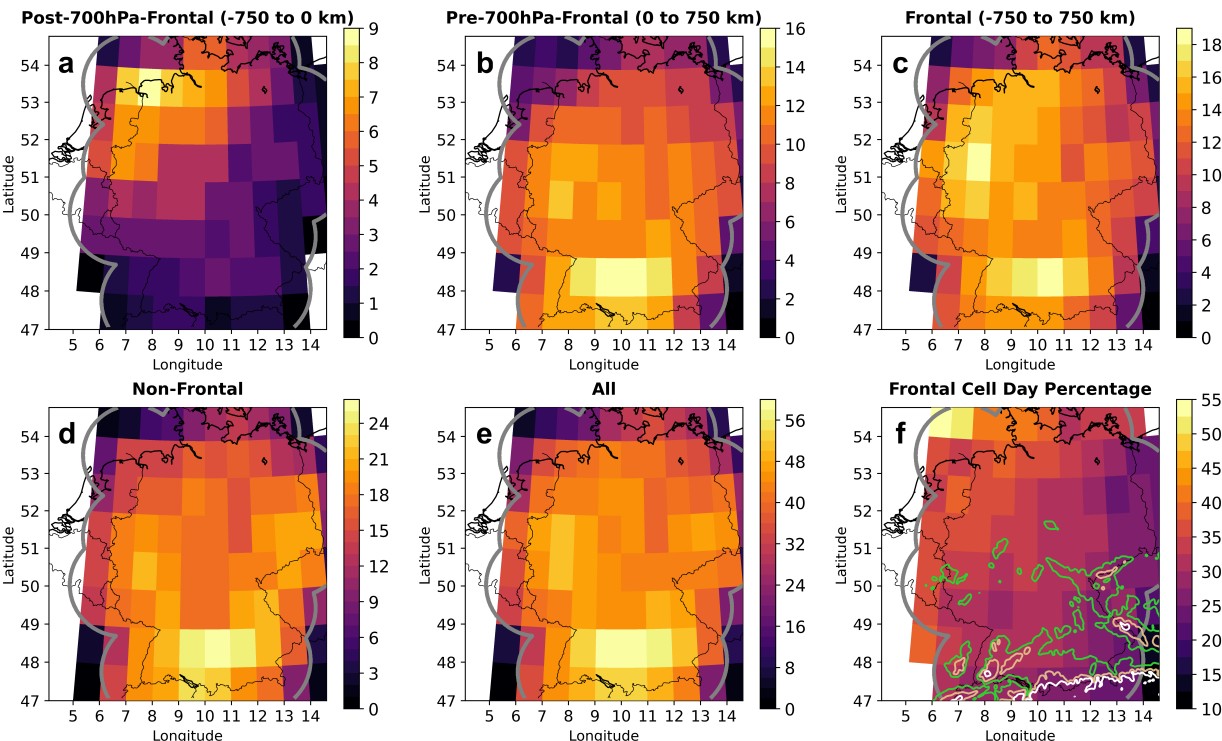

**Figure 8.** Number of convective cell days per warm season for 1 x 1 degree grid boxes for post-700hPa-frontal cells (a), pre-700hPa-frontal cells (b), cold-frontal cells (c), non-cold-frontal cells (d), all cells (e). Panel f shows the percentage of convective cell days associated with cold fronts (panel c divided by panel e) and the green, brown and white contours represent 500, 1000 and 1500 metres elevation, respectively. Note the different colorbar levels for each subplot.

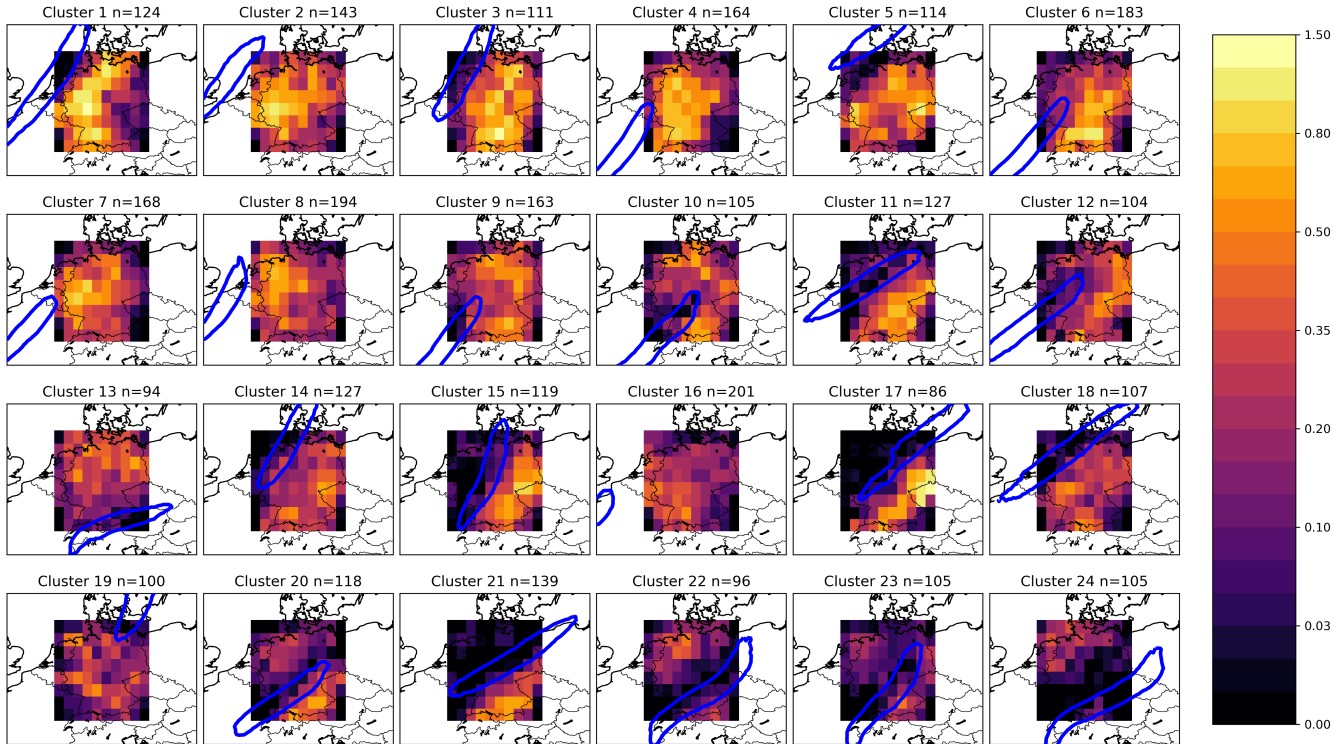

**Figure 9.** Clustered fronts using the k-means clustering algorithm in the domain [40N–60N,0–20E]. Subplot header represents the number of timesteps (n) contained in each cluster. The colorbar represents the number of convective cells per 1 x 1 degree grid box per timestep. The blue contours indicate where 50% of timesteps in the respective cluster have the front located at that grid point.

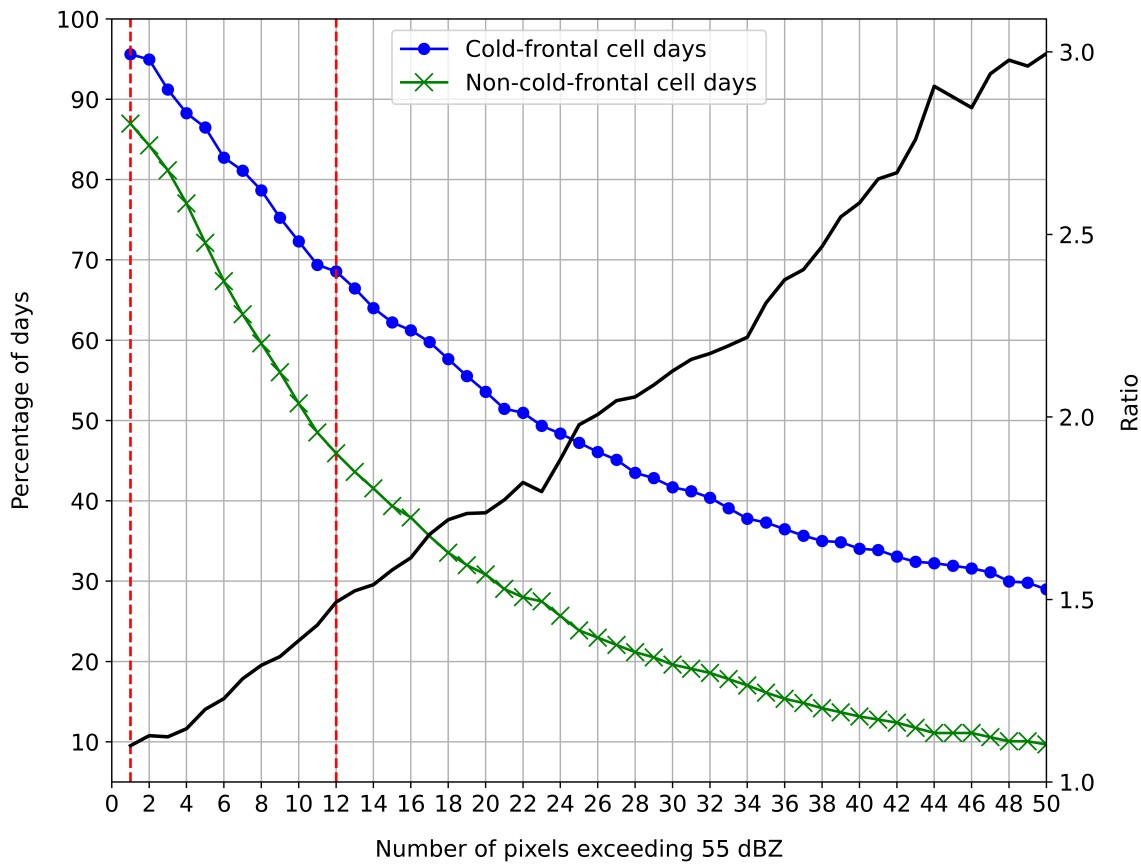

**Figure 10.** Percentage of days with at least one cell with n pixels exceeding 55 dBZ for cold-frontal cell days (blue) and non-cold-frontal cell days (green). The ratio (black) of frontal to non-cold-frontal cell days is shown on the secondary axis. The first dashed vertical red line represents the hail flag 1 and second dashed vertical line represents the hail flag 2 (excluding the 1 pixel exceeding 60 dBZ criterion).

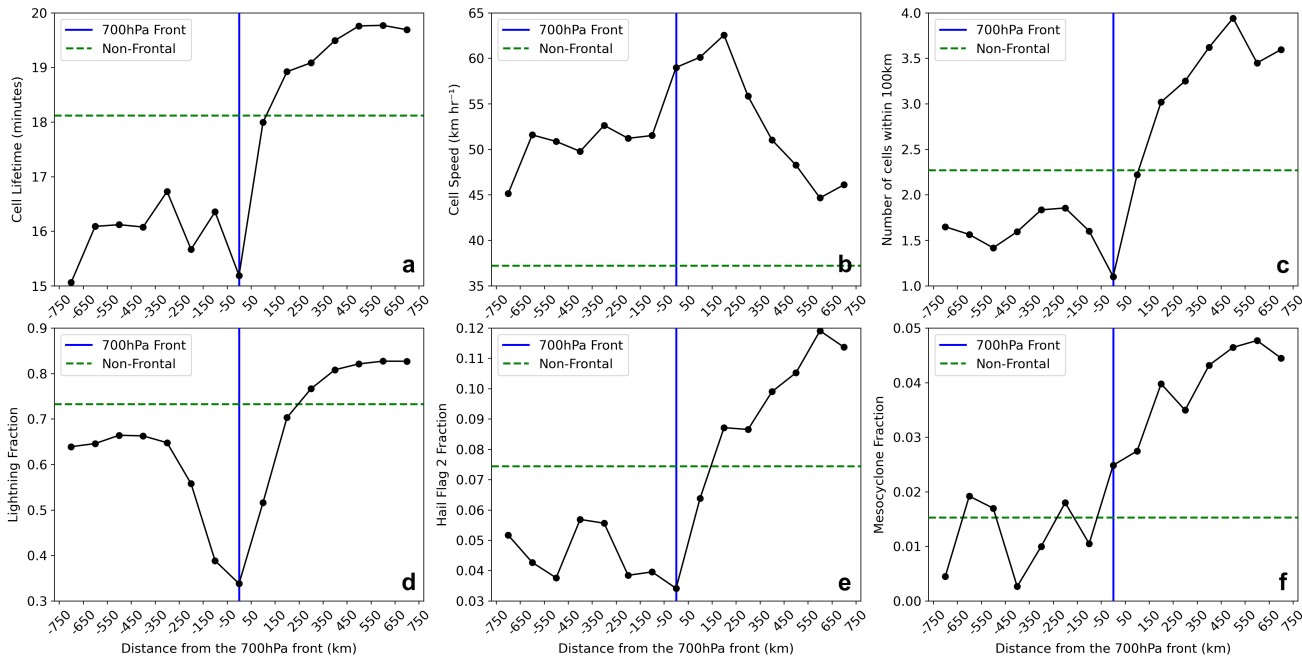

**Figure 11.** Mean cell lifetime (a), mean cell speed (b), mean number of cells within 100 km of cell centre (c), fraction of cells with lightning (d), fraction with hail flag 2 (e) and fraction with mesocyclones (f) depending on the cell-front distance (km). The dashed green line represents the non-cold-frontal cells fraction for reference.

| Name | Definitions and thresholds |
|---|---|
| Convective Cell | 15 pixels exceeding 46 dBZ |
| Hail Flag 1 | 1 pixel exceeding 55 dBZ |
| Hail Flag 2 | 12 pixels exceeding 55 dBZ or 1 pixel exceeding 60 dBZ |
| Lightning | 3 or more strikes 20 km or less from the cell centre during the cell's lifetime |
| Mesocyclone | 5 km equivalent diameter, 3 km depth, $10^{-3}$ s$^{-1}$ azimuthal shear or 15 ms$^{-1}$ rotational velocity |

**Table 1.** Convective cell, hail flag 1, hail flag 2, lightning and mesocyclone definitions. Since the KONRAD dataset has a spatial resolution of 1 km, 1 pixel ~ 1 km$^2$.

| Name | Cold-frontal cell days | Non-cold-frontal cell days | All cell days |
|---|---|---|---|
| Cell Count | 163,255 | 94,963 | 258,218 |
| Day Count | 614 | 775 | 1,389 |
| Cells per day | 265.9 | 122.5 | 185.9 |

**Table 2.** Summary of convective cell datasets. A cold-frontal cell day is defined as one where at least one convective cell was detected within 750 km of the 700 hPa cold-frontal line, otherwise the day is classified as a non-cold-frontal cell day.

## Appendix A:  Lightning climatology

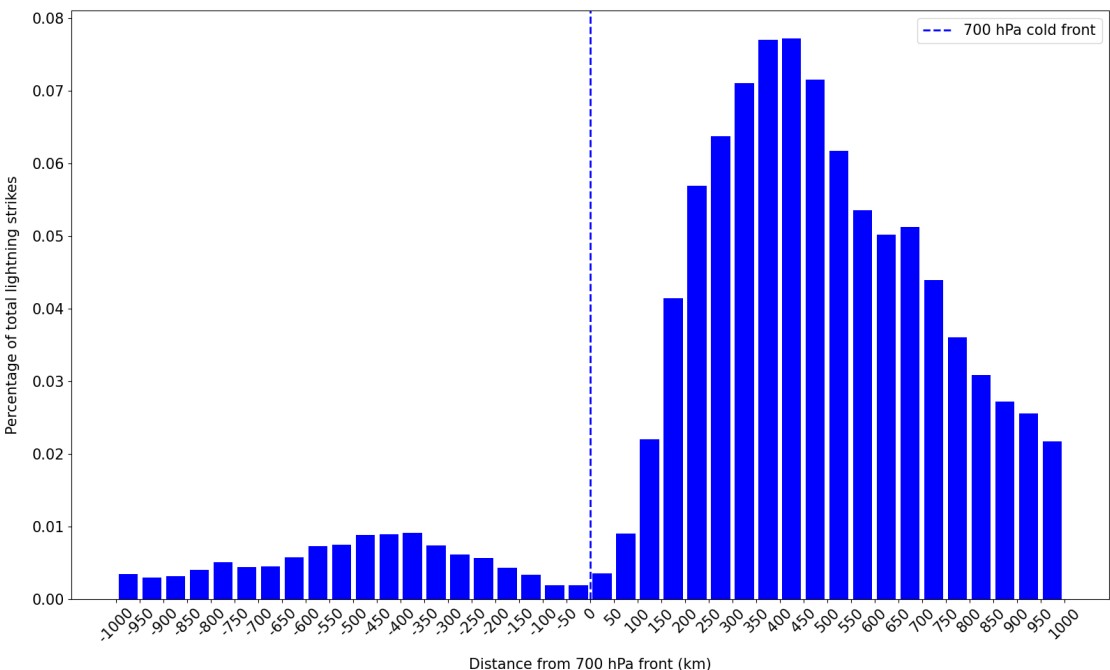

**Figure A1.** Lightning strike frequency for Germany depending on the distance from the 700 hPa front between 2010–2016 (April–September). Lightning data were provided by the Met Office, which uses an arrival time difference network (ATDnet) to detect lightning strokes (Met Office, 2020).

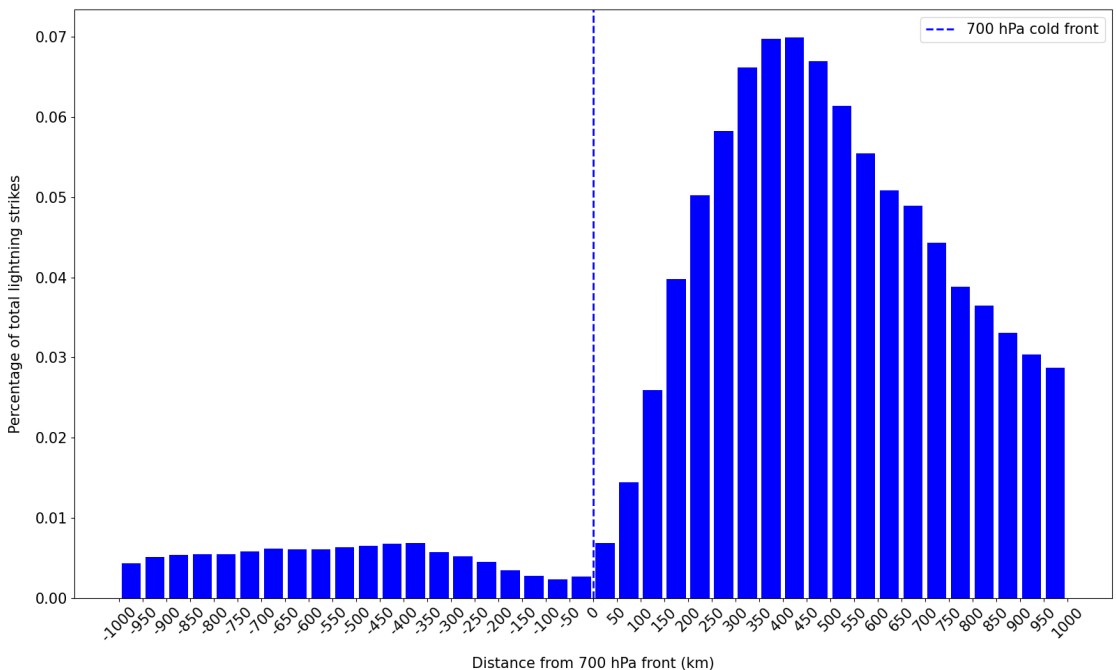

**Figure A2.** Lightning strike frequency for a sub-European domain (see grey domain in Figure 1) depending on the distance from the 700 hPa front between 2010–2016 (April–September). Lightning data were provided by the Met Office, which uses an arrival time difference network (ATDnet) to detect lightning strokes (Met Office, 2020).

**Appendix B: Randomisation Test**

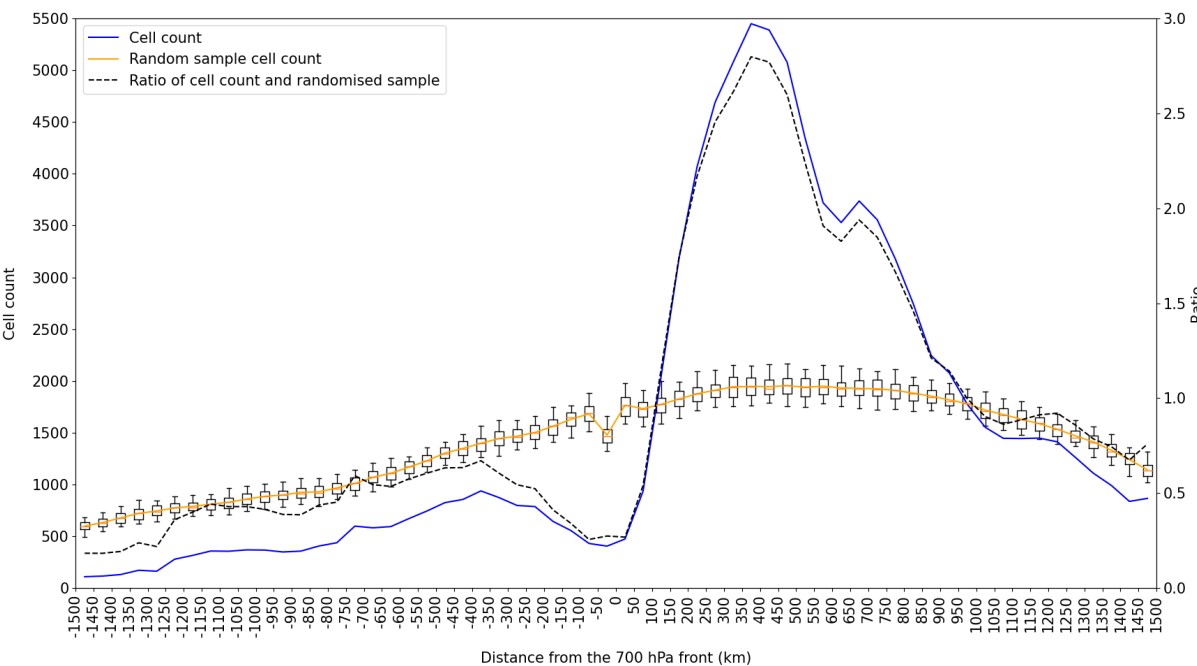

**Figure B1.** Cell count (blue), random sample cell count mean (orange) and ratio of cell count and randomised sample (black) on the secondary axis. The random sample was performed 100 times and the 5th and 95th percentiles are shown by the whiskers of the boxplots.

# Appendix C:  Clustering

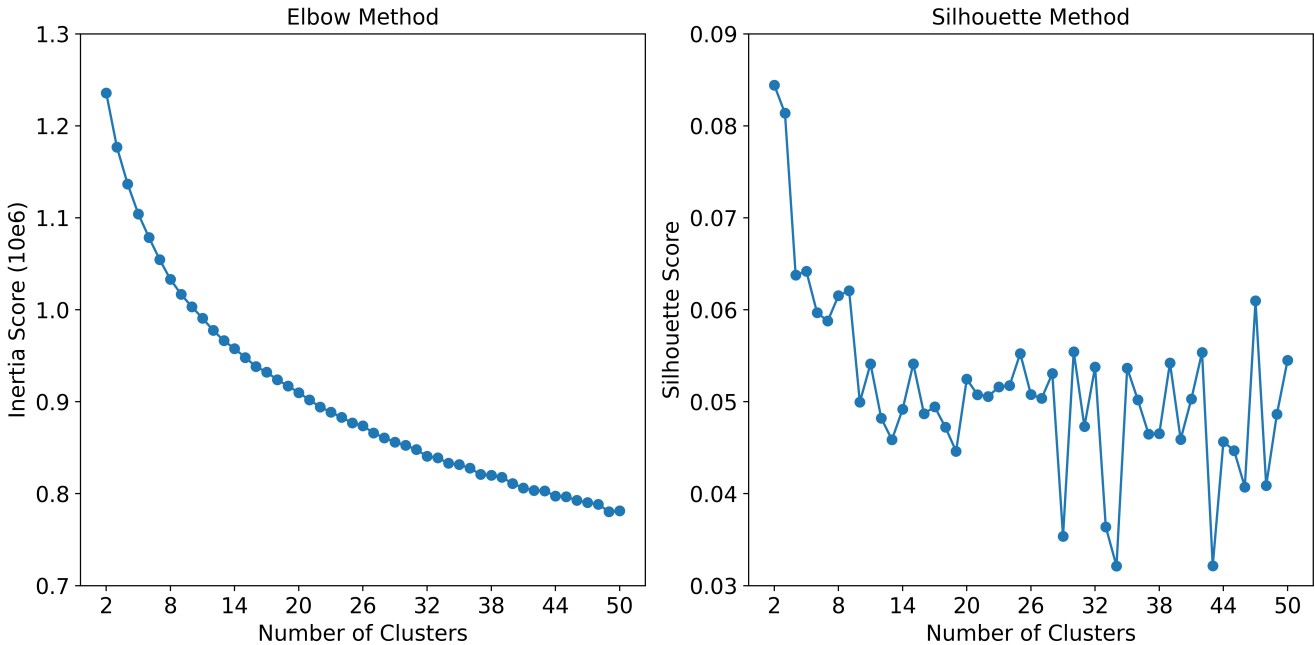

**Figure C1.** Elbow method (left) and silhouette score method (right) applied for cluster numbers between 2 and 50.

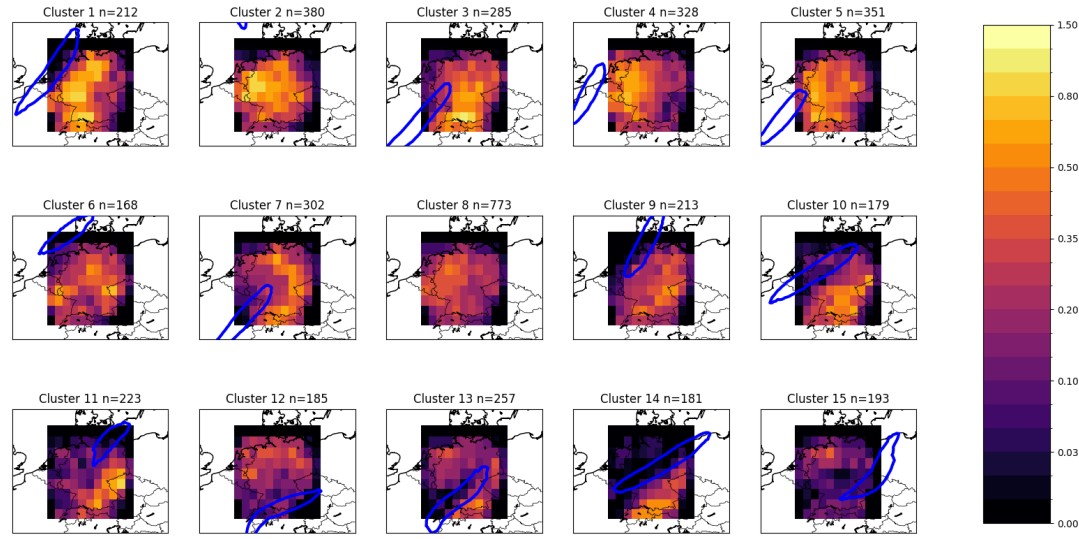

**Figure C2.** As Figure 9 but for 15 clusters. Absence of a blue contour indicates high within-cluster variance and no common front type associated to that cluster.

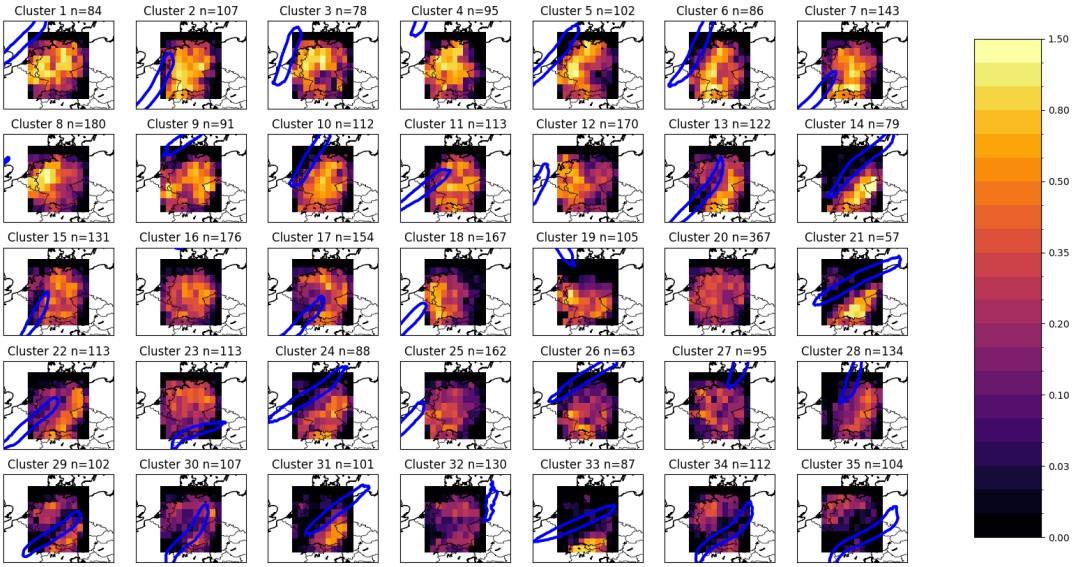

**Figure C3.** As Figure 9 but for 35 clusters. Absence of a blue contour indicates high within-cluster variance and no common front type associated to that cluster.

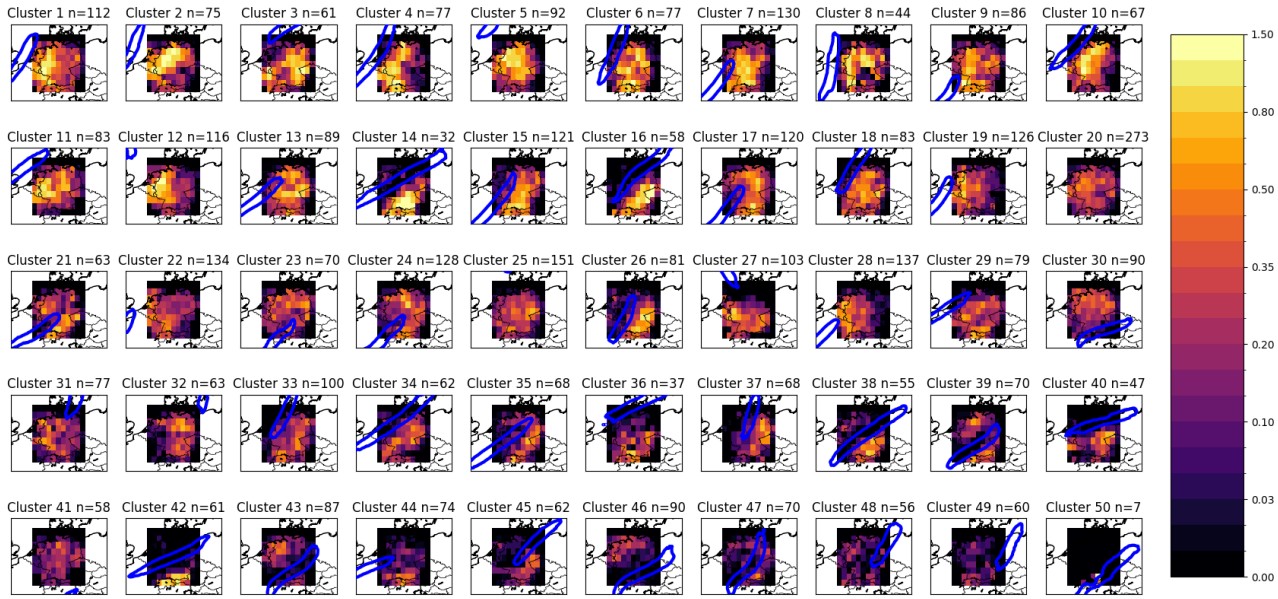

**Figure C4.** As Figure 9 but for 50 clusters. Absence of a blue contour indicates high within-cluster variance and no common front type associated to that cluster.

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
