# Peer review of "The climatology and nature of warm-season convective cells in cold-frontal environments over Germany"

_Natural Hazards and Earth System Sciences, 2023_

## Referee Comment (RC2)

**Review on "The climatology and nature of warm-season convective cells in cold-frontal environments over Germany" by Pacey et al.**

This study presents a climatology of convective cells associated with cold fronts in a front relative coordinate frame and compares these to convective cells occurring in non-cold frontal environments. Convective cells are shown to be much more frequent on cold front days than non-cold front days and the most likely location for convective cells to develop on cold front days is found to be 350 – 400 km ahead of the front. Overall, the manuscript is clear, well written and the results are supported by evidence. However, I have three major concerns regarding this manuscript.

The first major comment concerns how well this study fits the scope of this journal and the broader context of the results. In the manuscript the link to actual hazards is weak and little emphasis is given to this aspect. Lightning is considered but relatively briefly. It should be clearer how the results of this study inform about meteorological hazards. This study also only focuses on Germany, a choice which is motivated by the availability of radar data. While the authors do state that the study should be expanded to all of Europe, the current manuscript may be of limited interest to readers from other places than Germany. At a minimum the authors should attempt to address the question of how do these results apply to elsewhere in the world? Do they only apply over continental areas for example?

The second major comment regards the assumption that the surface front is 300 km ahead of the 700-hPa front. This is an oversimplification and likely is not accurate in many cases. Specific points:
1. It is stated that this assumption is based on ERA5 data, but this is not presented – it should certainly be shown even if only as supporting material.
2. It is stated that the surface convergence zone is 300 km ahead of the 700-hPa front. In some cold frontal cases the wind shift (i.e. convergence zone) is not co-located with the temperature gradient so it may not be the thermal gradient which is 300 km ahead of the 700-hPa front. It is also inconsistent to use convergence to locate the surface front but a thermal gradient for the 700-hPa front.
3. This simple approach does not consider kata-cold fronts in which the front appears to slope forward with height as the cold air aloft has overrun the surface front. Kata cold fronts can certainly trigger elevated convection and these fronts should be considered separately.

The third major comment concerns the clustering presented in section 3.3.3. Specific issues here are:
1. The manuscript lacks details on exactly how the clustering was done (e.g. was any normalisation on the input features performed?).
2. The choice of the number of clusters appears subjective whereas the silhouette score and elbow plots could be used to better justify the final number of clusters.
3. The number of clusters (30) is too large to be of practical use to e.g., forecasters
4. The justification for removing 6 of these 30 clusters is not clear and it appears that the clustering has identified 6 clusters which are not physically consistent – strongly suggesting that the clustering has not been performed in an optimal manner.
5. The only outcome of the clustering that is presented is the number of cells and the location of the front. It would be helpful for forecasters to also see additional meteorological variables associated with each of these clusters, for example, the MSLP and equivalent potential temperature.

Below I also list some minor comments which would certainly improve the manuscript:

Minor comments

1. Line 41 – 42. Please expand this sentence to make it clearer. It needs to be stated that this is due to the frontal surface sloping rearwards with height.
2. Line 62, Question 3. This could be written in a manner so it can stand alone and does not need a reader to refer back to Q1 and Q2. This would likely make it clearer and easier to understand.
3. Line 125. Figure 1. Could the domain where fronts are identified in be marked on this figure?
4. Section 2.1.1. The criteria used to identify the fronts are quite large so will only identify quite strong fronts in terms of the thermal gradient. Do the results depend on these thresholds, and in particular, do the fronts still hold if weaker fronts are also considered? If not, it should be stressed more clearly that these results only apply to strong cold fronts.
5. Section 2.3. How were these four examples selected and how representative of the whole data set are they? They look like quite standard fronts, so I am wondering if the method works well with more complicated or less uniform fronts.
6. Section 2.3 / method. "Timesteps containing two or more cold front lines in the domain were omitted". Since timesteps with two fronts present are omitted, this means that this method only works for a small area and could not be expanded to e.g., European scale (even if the radar data was available). This is a notable limitation of this method which should be clearly highlighted, or the method improved to allow two or more fronts to be present at the same timestep.
7. Related to the point above, neglecting timesteps with two or more fronts present means that double fronts will be automatically ignored which may add a systematic bias to the results.
8. Line 176. There is a typo here "in In Figure 2a…"
9. Section 3.1. There are a lot of numbers, percentages especially at the start of this section and it is hard to read. Many of these numbers etc, are in Table 1, but Table 1 is not referred to much here. It would help a reader to refer to Table 1 more. Furthermore, this section may be clearer if the number of cold front days was discussed first, and this information was added to table 1.
10. Line 199, should this be a comma before All rather than a period?
11. Line 210. Is the surface convergence influenced by land use, coastlines, topography etc.?
12. Line 218 and elsewhere after this the phrase "pre-700-frontal environment…" is used. Should hPa be added after 700 here?
13. Line 272 – 274. How does this spatial climatology of convective cells relate to the spatial climatology of fronts as shown in Figure 1? Adding a sentence to relate these aspects would be helpful for a reader.

---

## Author Comment (AC2)

*Authors' responses in red italics*

**RC2**

**Review on "The climatology and nature of warm-season convective cells in cold-frontal environments over Germany" by Pacey et al.**

This study presents a climatology of convective cells associated with cold fronts in a front relative coordinate frame and compares these to convective cells occurring in non-cold frontal environments. Convective cells are shown to be much more frequent on cold front days than non- cold front days and the most likely location for convective cells to develop on cold front days is found to be 350 – 400 km ahead of the front. Overall, the manuscript is clear, well written and the results are supported by evidence. However, I have three major concerns regarding this manuscript.

*We thank the reviewer for taking the time to review the manuscript and for their constructive feedback. The revised manuscript will include several changes based on the reviewer's comments which are outlined in more detail below under each specific comment.*

The first major comment concerns how well this study fits the scope of this journal and the broader context of the results. In the manuscript the link to actual hazards is weak and little emphasis is given to this aspect. Lightning is considered but relatively briefly. It should be clearer how the results of this study inform about meteorological hazards. This study also only focuses on Germany, a choice which is motivated by the availability of radar data. While the authors do state that the study should be expanded to all of Europe, the current manuscript may be of limited interest to readers from other places than Germany. At a minimum the authors should attempt to address the question of how do these results apply to elsewhere in the world? Do they only apply over continental areas for example?

*The study focuses on convective cells which are known to be linked to hazards such as lightning, wind, rain and hail. From Figure 10d we see that most cells are indeed associated with lightning; around 80% of cells in the warm-sector. Furthermore, in the final section of the paper (Section 3.4) we go into extensive detail regarding the links to hazards e.g., intensity of cells and mesocyclone frequency in the cold-frontal framework. We will however add additional discussion in the introduction about the motivation of this work being related to convective hazards. We will also emphasise in the conclusion that this work improves understanding of convective hazard climatology.*

*Regarding the application to other parts of Europe we found that lightning frequency has the same distribution around the 700 hPa front in a larger European domain (see Figure R1). This gives us an early indication that the results could apply on a broader European scale but since radar data (especially doppler wind velocities which are needed for mesocyclone detection) is not readily available on a European scale it is not feasible to reproduce the entire results on a European scale at this time. We will include some discussion regarding our preliminary results on lightning frequency in a larger European domain in the new manuscript.*

[Figure]

*Figure R1: Lightning stroke count depending on the distance from the 700 hPa front between 2007–2016 (April–September). Lightning data were provided by the Met Office, which uses an arrival time difference network (ATDnet) to detect lightning strokes (Met Office, 2020).*

The second major comment regards the assumption that the surface front is 300 km ahead of the 700-hPa front. This is an oversimplification and likely is not accurate in many cases. Specific points:

1. It is stated that this assumption is based on ERA5 data, but this is not presented – it should certainly be shown even if only as supporting material.
2. It is stated that the surface convergence zone is 300 km ahead of the 700-hPa front. In some cold frontal cases the wind shift (i.e. convergence zone) is not co-located with the temperature gradient so it may not be the thermal gradient which is 300 km ahead of the 700-hPa front. It is also inconsistent to use convergence to locate the surface front but a thermal gradient for the 700-hPa front.
3. This simple approach does not consider kata-cold fronts in which the front appears to slope forward with height as the cold air aloft has overrun the

surface front. Kata cold fronts can certainly trigger elevated convection and these fronts should be considered separately.

*We thank the reviewer for their feedback regarding the surface front location relative to the 700 hPa front. The assumption is not only based on the mean surface convergence in ERA5 but also on knowledge of cold frontal slopes (1:100). We do not claim the surface front is always located exactly 300 km ahead of the 700 hPa front rather this is where the convergence is highest climatologically speaking. For this paper we are primarily interested in cold-frontal convective cell climatology. We are not focusing on specific case studies where, as the reviewer rightly mentions, the surface front location relative to the 700 hPa front may vary.*

1. *We will include the climatological convergence at different height levels in the 700 hPa framework in the updated manuscript as the reviewer suggests (see Figure R2). The climatological value of CAPE, surface dewpoints and surface shortwave radiation are also overlayed.*

[Figure]

*Figure R2: Climatological convergence between 975 hPa and 500 hPa (shaded), MUCAPE (dashed line), surface dewpoint (straight line), surface shortwave radiation (straight dotted line). Excluding convergence, the climatological values of the variables are normalised between 0 and 1 so the distribution around the front can be compared. Variables were derived in ERA5 data.*

2. *Since the surface convergence is relevant for convective initiation, we believe this is an appropriate reference point. Furthermore, the surface temperature gradient is*

*typically not as well-defined, especially in the warm-season. For this reason, automatic front detection methods are usually applied above the boundary layer.*

3. *The thermodynamic gradient of kata fronts still backs with height as with an ana front (Moore and Smith, 1989, their Figure 1), thus the approach holds for both types of fronts. Furthermore, if these upper-level humidity fronts were being detected using our cold-front detection methods we would expect to find a much larger cell frequency behind the 700 hPa front. Since Kata fronts typically evolve from Ana fronts, it is not trivial to separate such fronts on a case-by-case basis. It may however be interesting for future work to investigate the effects of the overrunning upper-level dry intrusion on the resulting convection when a kata cold front is present. It is expected this increases the strength of the capping inversion thus inhibiting the premature release of convection which is relevant for severe convection. In the revised manuscript we will mention that we do not explicitly consider ana and kata cold fronts separately, but our results suggest we do not detect many of these upper-level humidity fronts in our algorithm.*

The third major comment concerns the clustering presented in section 3.3.3. Specific issues here are:

1. The manuscript lacks details on exactly how the clustering was done (e.g. was any normalisation on the input features performed?).
2. The choice of the number of clusters appears subjective whereas the silhouette score and elbow plots could be used to better justify the final number of clusters.
3. The number of clusters (30) is too large to be of practical use to e.g., forecasters
4. The justification for removing 6 of these 30 clusters is not clear and it appears that the clustering has identified 6 clusters which are not physically consistent – strongly suggesting that the clustering has not been performed in an optimal manner.
5. The only outcome of the clustering that is presented is the number of cells and the location of the front. It would be helpful for forecasters to also see additional meteorological variables associated with each of these clusters, for example, the MSLP and equivalent potential temperature.

1. *We thank the reviewer for their feedback on the clustering. We agree that further details would be useful in this section. For the reviewer's information, no normalisation was carried out on the input data as it is binary, each grid point is either 1 for a front grid point or 0 for a non-front grid point. Such information will be added to the revised manuscript for clarity.*

*2. The reviewer is right that the elbow method or silhouette score can be used to select the cluster number in a more objective manner. However, no clear elbow was identified using the elbow method and no optimal silhouette score (see Figure R3). In this case user expertise was used instead of the metrics. Further justification of the cluster number choice is discussed in the responses below. We will add a sentence explaining that the elbow method and silhouette scores were considered before selecting the cluster number and include Figure R3 below in the Appendix.*

[Figure]

*Figure R3: Elbow method (left) and silhouette score method (right) applied for cluster numbers between 2 and 50.*

*3. We do not think that the cluster size is too large, forecasters are aware that cold fronts have a variety of orientations and different geographical locations. For example, forecasters at the German Weather Service (DWD) are used to identifying 29 different synoptic weather patterns (Wapler and James, 2014, their table 1). We carried out extensive testing between 10 and 50 clusters finding that lower cluster numbers resulted in fronts with different features being grouped in the same cluster. Larger cluster numbers on the other hand had less variance in each cluster but the results become less interpretable. Ultimately, a compromise somewhere must be found.*

*4. 6 clusters were removed due to high variance within those clusters. This however does not indicate the clustering has not been performed in an optimal manner. When using a higher cluster number such as 50 these 6 clusters would have been split into separate clusters with less variance in the cluster, but as mentioned at 50 clusters it is hard for the reader to interpret such results.*

*5. The primary focus of this analysis is to see how the orientation/position of the front is linked to the cell detections. A composite of the equivalent potential temperature*

*field for each cluster will only show a boundary where the front is located, thus not adding any additional information.*

Below I also list some minor comments which would certainly improve the manuscript:

Minor comments

1. Line 41 – 42. Please expand this sentence to make it clearer. It needs to be stated that this is due to the frontal surface sloping rearwards with height. *On L39 we already mention the cold-front slope, we will amend this line to read "Due to the rearward slope of cold fronts, there is no concrete...". Thank you for the suggestion.*

2. Line 62, Question 3. This could be written in a manner so it can stand alone and does not need a reader to refer back to Q1 and Q2. This would likely make it clearer and easier to understand. *We think this would lead to a rather long research question. The reader would have just read questions 1 and 2 so it will be clear what is meant by question 3, we prefer to avoid unnecessary repetition in this case.*

3. Line 125. Figure 1. Could the domain where fronts are identified in be marked on this figure? *The domain showed in Figure 1 is the domain in which fronts were detected. We will add a sentence on this in the figure caption in the revised manuscript so it is clear. Thank you for the suggestion.*

4. Section 2.1.1. The criteria used to identify the fronts are quite large so will only identify quite strong fronts in terms of the thermal gradient. Do the results depend on these thresholds, and in particular, do the fronts still hold if weaker fronts are also considered? If not, it should be stressed more clearly that these results only apply to strong cold fronts. *We will emphasise in the method section (section 2.3) that we are focusing on cases where a single large-scale cold front is present in western/central Europe, and that smaller-scale and weaker fronts (below 6K / 100km) are not explicitly considered. Thank you for the comment.*

5. Section 2.3. How were these four examples selected and how representative of the whole data set are they? They look like quite standard fronts, so I am wondering if the method works well with more complicated or less uniform fronts. *The examples were selected to show fronts with different orientations and with cells in different locations relative to the front. The clustering analysis shows the primary frontal types contained in the dataset. The study does indeed focus on standard fronts where only one large-scale cold front is present over*

*western/central Europe. We will include this point after the "timesteps containing two or more cold fronts" comment.*

6. Section 2.3 / method. "Timesteps containing two or more cold front lines in the domain were omitted". Since timesteps with two fronts present are omitted, this means that this method only works for a small area and could not be expanded to e.g., European scale (even if the radar data was available). This is a notable limitation of this method which should be clearly highlighted, or the method improved to allow two or more fronts to be present at the same timestep. *In the event there are two large-scale fronts (L~1000 km) present it is not trivial to decide which front to use for the cell-distance calculations. Selecting the largest front for example may induce a bias as the smaller front may also have some influence on the resulting convection. As mentioned in the last comment we will remark in the revised manuscript that we focus on cases with a single large-scale (L~1000 km) cold front present over western/central Europe.*

7. Related to the point above, neglecting timesteps with two or more fronts present means that double fronts will be automatically ignored which may add a systematic bias to the results. *We believe the opposite is true as when two large-scale fronts are present it is not clear which front should be selected without inducing a systematic bias.*

8. Line 176. There is a typo here "in In Figure 2a..." *Thank you, this will be corrected.*

9. Section 3.1. There are a lot of numbers, percentages especially at the start of this section and it is hard to read. Many of these numbers etc, are in Table 1, but Table 1 is not referred to much here. It would help a reader to refer to Table 1 more. Furthermore, this section may be clearer if the number of cold front days was discussed first, and this information was added to table 1. *We agree that it would be useful to refer to Table 1 earlier in the section, thank you for the suggestion. The second line can be removed as the information is already contained in the table, thus making the text shorter.*

10. Line 199, should this be a comma before All rather than a period? *We feel including this in the same sentence would make it too lengthy. Instead, we will revise it to read "Such synoptic patterns are likely..."*

11. Line 210. Is the surface convergence influenced by land use, coastlines, topography etc.? *ERA5 at ~25 km resolution may struggle to resolve such features and when averaging across several thousand timesteps such effects will be filtered out leaving the surface convergence due to the front.*

12.Line 218 and elsewhere after this the phrase "pre-700-frontal environment…" is used. *We will use pre-700hPa-frontal and post-700hPa-frontal throughout in the revised manuscript for consistency, thank you for bringing this to our attention.*

Should hPa be added after 700 here? *We will use the notions of pre-700hPa-frontal and post-700hPa-frontal throughout.*

13.Line 272 – 274. How does this spatial climatology of convective cells relate to the spatial climatology of fronts as shown in Figure 1? Adding a sentence to relate these aspects would be helpful for a reader. *North-west Germany has the largest fraction of cell days on cold-frontal cell days which could indeed be linked to the high frequency of fronts seen in this region. We will add discussion regarding this in section 3.3.2 of the revised manuscript. Thank you for the comment.*

*References*

*Moore, J. T., and K. F. Smith, 1989: Diagnosis of Anafronts and Katafronts. Wea. Forecasting, **4**, 61–72, https://doi.org/10.1175/1520-0434(1989)004<0061:DOAAK>2.0.CO;2.*

*Met Office, 2020: Lightning strike location data. Met Office, 3 pp., https://www.metoffice.gov.uk/binaries/content/assets/metofficegovuk/pdf/data/adtnet_data_sheet.pdf.*

*Wapler, K. and James, P, 2014: Thunderstorm occurrence and characteristics in Central Europe under different synoptic conditions, Atmospheric Research, 158, https://doi.org/10.1016/j.atmosres.2014.07.011.*

---

## Author Comment (AC3)

***Authors' responses in red italics***

**RC3**: 'Comment on nhess-2023-39', Anonymous Referee #3, 05 May 2023  reply

**General comments:**

This study analyses the location and characteristics of deep moist convection associated with cold fronts over central Europe. Many novel insights are gained and nicely embedded in the existing literature. Overall, the methods, structure, and the figures are of a high quality. I don't have any reasons for rejection and my comments can mostly be seen as suggestions, although I agree with some of the concerns of reviewer #2 (see major comment 3 below).

*We thank the reviewer for taking the time to review the manuscript and for their constructive feedback. The revised manuscript will include several changes based on the reviewer's comments which are outlined in more detail below under each specific comment.*

**Major comments:**

1) line 132: The bias in Fig. 1 looks more than "slight". If it were a weak bias, shouldn't more fronts be expected towards the Atlantic where strong lows are originating from? Or is the theta gradient increasing over land? I think some more discussion for why the dataset is still useful for your purpose seems warranted, e.g., that you are mostly interested in fronts with convection impacting central Europe.

*We will revise the manuscript indicating there is a bias towards the NW and SE of the domain and since we focus on Germany this does not affect the results, thank you for the suggestion. However, we note that several other front climatologies also did not find the highest front frequency in the Northern Atlantic e.g., Fig. 5c in Schemm et al. (2015) and Figure 7 in Niebler et al. (2022).*

2) 159 Do you think any biases resulted from only counting the first cell detection? I think the approach is good enough, but I could imagine that long-lived cells change their location relative to the front over time?

*Figures 4 and 5 in the current manuscript were also created using cells at all detection times and no significant differences were observed. From Figure 10a and Figure 10b in the current manuscript we see that cells have a typical lifetime between 15-20 minutes and propagate at a speed of between 45–65 km hr$^{-1}$ in cold frontal environments. The mean distance a cell propagates is therefore between 11–21km. We produce the results in 50 km intervals, supporting why we see no significant differences by including cells at later detection times. The*

*primary motivation for only counting the first detection was to remove duplicate counting of cells.*

3) Perhaps my main point of criticism (or the aspect of the study which could be improved the most) is that although you discuss lift mechanisms ahead and at the front, not much analysis is done to locate these features relative to the 700hPa front location. I realize that this is not easy and changes a lot from case to case, but since the study claims to describe the "nature of cold-frontal convection", a deeper analysis seems warranted. For instance, is it possible to add the locations of the surface front and pre-frontal convergence relative to the 700hPa front in Figs. 4 and 5 (or rather the range of locations in your dataset, e.g., as a box and whisker)? Could this be estimated from the ERA5 data you used? An alternative would be to go through some cases manually and indicate these boundary positions for each case in the plots.

*We thank the reviewer for their feedback. We will include the figure below (Figure R1) which shows the climatological convergence at different pressure levels as a function of the distance from the 700 hPa front. This shows the typical lift at different distances from the front and at different height levels. This was originally planned to be left for a future publication, but we agree this analysis would be useful for readers. The climatological value of CAPE, surface dewpoint and solar radiation are also overlayed.*

[Figure]

*Figure R1: Climatological convergence between 975 hPa and 500 hPa (shaded), MUCAPE (dashed line), surface dewpoint (straight line), surface shortwave radiation (straight dotted line). Excluding convergence, the climatological values of the variables are normalised between 0 and 1 so the distribution around the front can be compared. Variables were derived in ERA5 data.*

4) 295 Looking at the number of cells might be misleading because larger storm systems (MCS) result in less cells counted even though they have a larger impact. You don't need to change the analysis but this should be made clear again to the reader. In general, strengths and weaknesses of the KONRAD dataset are not discussed much.

Another example would be that you speculate that the pre-frontal diurnal cycle is broader because of more MCS. Wouldn't that also mean that less individual cells were detected there (because one MCS has larger but less cells)? The opposite is seen in Fig. 10c. Is that because of flaws in the detection algorithm?

*We have seen from a few case study examples that MCS's are typically associated with a larger number of cells. There are two explanations for this, firstly a continuous line exceeding 46 dBZ may not always be present, therefore several cells are detected within an MCS. Secondly, due to cell recycling in an MCS, KONRAD will likely detect new cells at subsequent timesteps. See an example case below (Figure R2) with 4 radar timesteps shown (15-minute interval) and KONRAD cell detections at 5-minute intervals (bottom) for a case study in August 2013. We will not include these figures in the manuscript but we will include additional discussion about cell detections in KONRAD during a typcial MCS event.*

[Figure]

[Figure]

[Figure]

*Figure R2: OPERA Radar Data showing max dBZ (Huuskonen et al. 2014; Saltikoff et al. 2019) (top). KONRAD cell detections with first detections in blue and secondary (or later) detections in orange (bottom).*

**Minor comments and technical corrections:**

24-25 An alternative to this explanation would be varying DMC ingredients (e.g., CAPE) in different regions along the front. *We will add a line that reads "Furthermore, the importance of difference mechanisms may vary across the front, e.g., stronger synoptic lift near the front but increases in solar heating away from the front." Thank you for the suggestion.*

26-38 There is a lot of good content here, but the text seems a bit unstructured. For instance, pre-frontal convergence lines are also a result of an ageostrophic circulation (Dahl and Fischer 2016), not only the lift at the front. Also, you could make a bit clearer that you start discussing mechanisms ahead of the front, then at the front and then behind, for instance by ending the first sentence (l. 27) with "... location relative to the

front." And then starting the next sentence with "Ahead of the front, …". *We will add that pre-frontal convergence lines are also a result of ageostrophic circulation, thank you for the suggestion. Indeed, the sentence will be clearer by saying"…. depending on the convective initiation location relative to the front". Thank you.*

41-42 I'm not quite sure if I understand the point. If this holds true for a surface observer, why is it not true? Are you referring to the fact that the convective cloud is not necessarily in the same location as the heaviest precipitation? *We are pointing out that if convection is observed shortly after the passage of the surface front, the saturated cloud region is largely on the warm-side of the front thus is not post-frontal. Indeed, the precipitation falls to the ground into the cooler post-frontal airmass, but the cell itself is typically on the warm-side of the front.*

97-98 If I understand correctly, "higher" would only be true for cold fronts, because for warm fronts v_f would be negative. Did you mean to say the magnitude of v_f is higher with stronger advection? *Yes, thank you. We will revise the manuscript accordingly.*

101-104 Remains unclear how L was determined. Is it a continuous length of points where the other criteria were met? What about brief gaps in the boundary detection? *The coordinates of where TFP=0 are located using interpolation. The distance from each adjacent point was calculated and summed for the whole line. In the event there is a gap this is considered a separate feature and not added to the total length. We will expand on our current explanation in L101–102 including the information mentioned above.*

141 "At the start of this study," (comma in English after such introductory words for a sentence" *Thank you.*

section 2.2 in general: Just a suggestion, but I would bring in the hail, lightning, and mesocyclone detection methods later when they are needed. Jumping back and forth between the different dBZ thresholds was a bit confusing here. *We did consider this, but we felt it would disrupt the flow of the results to include such technical definitions in the results section. We will include the different thresholds in a table so it is easier for the reader refer back to.*

155 Make clear what "such" is referring to. I'm assuming you mean the hail, lightning and mesocyclone detections? *Thanks for the suggestion, indeed it is referring to the hail flag, lightning and mesocyclone definitions. We will revise the manuscript accordingly.*

163 comma behind "2.2". *Thank you.*

164 I like this numbering of criteria. Makes it really clear. *Thank you for the feedback.*

185 Comma after "September" *Thank you, we will make changes to the revised manuscript.*

187 and 262 One convective cell is a fairly low threshold. Days with >1 and >100 days are weighted equally with this method, correct? Perhaps discuss this caveat of considering cell days by e.g., showing a histogram of the number of cells per cell day to make clear that most cell days were really days with much convective activity?

*The one cell per day criterion is just to assign a day as a cold-frontal or non-cold-frontal cell day. For this analysis we are actually interested in the mean number of cells per day so by just selecting days with say >20 cells would add a bias to the results.*

189 Associated "with" *Thank you.*

239-240 There is a clear secondary maximum around 750 km ahead of the front in Fig. 5, which is interesting. Do you speculate that this is just free convection in the warm sector or pre-frontal convergence serving as another (weak) trigger? By the way, I really like this Figure. Is it necessary to have non-linear color scheme. It may look nicer but I think it makes it harder to interpret the numbers. If it's necessary, it should at least be mentioned in the caption. *From Figure 4 we see that between 600–750 km the cell frequency remains stable with a slight increase. It is plausible that this is linked to pre-surface-frontal convergence lines as they are usually found around 300km ahead of the surface front. We will add additional discussion in the updated manuscript, thank you for bringing this to our attention.*

*Since there are a lot more cells pre-700hPa-frontal compared to post-700hPa-frontal it is necessary to use a non-linear colorbar to highlight the post-700hPa-frontal diurnal cycle. We will note this in the figure caption, thank you for the suggestion.*

248-255 How do supercells fit into this picture? Their long lifetime could also lead to a broader diurnal cycle. You only mention Wapler (2016) briefly and without context. *We will add a line after the Wapler (2016) reference that says "This indicates that supercells, which typically have a longer lifetime, may also be linked to the observed weakened diurnal cycle". Thank you for the suggestion.*

259 First time reading this, I was confused what you mean by "vary". It might just be me, but if you refer to the spatial distribution, it's clearer to say something like: "The frequency of convective events varies in different parts of Europe." *Thank you for the suggestion, this will indeed make it clearer to readers. We will revise accordingly.*

268-270 Sounds like you believe this is due to a general increase towards the south / the mountains. Your resolution is fairly coarse, but the spatial distribution you observe

would also be consistent with mesoscale enhanced convective activity in local parts of Germany, e.g., South of Stuttgart (Piper 2017 Fig. 3, Kunz 2010). *Thank you for the reference, we will add this to the revised manuscript.*

295 Here and elsewhere: "Colorbar" *Thank you.*

303 This is consistent with enhanced activity in the Erzgebirge region (Piper Kunz 2017). *Thank you for the reference, we will add this to the revised manuscript.*

322 Such a pattern is also often associated with advection of an elevated mixed layer from Southwest Europe and pre-frontal convergence lines (Dahl and Fischer 2016). *Thank you for suggestion and reference, we will add these details to the revised manuscript.*

378-385 This last paragraph was a nice finish and the results are plausible. The conclusion section is also nice and precise. *Thank you for the feedback.*

Fig. 8 I also liked this Figure and analysis. Could you add the average number of cells per event over whole domain in the top of each subplot? Even though some clusters might be rare, their impact/number of cells might be large, so giving the reader information about the typical number of cells could be useful. *The number of cells per grid box is already normalised by the number of timesteps in the cluster accounting for rare cluster types.*

Fig. 10 Also very informative plot. Titles for each subplot would be helpful to avoid having to jump between caption and plot. *Thank you for the nice suggestion, we will recreate the figure for the new manuscript.*

*References*

*Huuskonen, A., E. Saltikoff, and I. Holleman, 2014: The operational weather radar network in Europe. Bull. Amer. Meteor. Soc., **95**, 897–907, https://doi.org/10.1175/BAMS-D-12-00216.1.*

*Schemm, S., Rudeva, I., and Simmonds, I.: Extratropical fronts in the lower troposphere–global perspectives obtained from two automated methods, 2015: Quarterly Journal of the Royal Meteorological Society, 141, 1686–1698, https://doi.org/10.1002/qj.2471*

*Saltikoff, E., and Coauthors, 2019: OPERA the Radar Project.* Atmosphere*, **10**, 320, https://doi.org/10.3390/atmos10060320.*

*S. Niebler, A. Miltenberger, B. Schmidt, and P. Spichtinger, 2022: Automated detection and classification of synoptic-scale fronts from atmospheric data grids. Weather and Climate Dynamics, 3(1):113–137. doi: 10.5194/wcd-3-113-2022. URL https://wcd.copernicus.org/articles/3/113/2022/*

---

## Author Response (AR1)

*Authors' responses in red italics*

**RC1**: 'Comment on nhess-2023-39', Anonymous Referee #1, 04 Apr 2023  reply
**General comments**

The submitted manuscript contains a climatological study of convective cells and mesoscale cyclones and their nature relative to a set of automatically detected near-surface cold fronts. This combination of established front detection, which provides information on large-scale flow, and a radar-based algorithm for detecting and tracking convective cells is unique and novel.

The paper is generally well written, and the illustrations are of high quality. Most parts of the paper are descriptive without going into much detail of the individual forcing mechanisms. Something the author may reserve for the future, but the manuscript could be expanded in this regard as well, for example, by adding more information on surface slope (e.g., orographic lifting in regions along the Black Forest, Harz, Rhön, for example).

The authors derive some interesting properties from their methods, such as cell speed and cell lifetime versus distance to the next cold front. The authors might consider adding more information about the relationship between these properties and front strength or front orientation (some are more zonal some are more meridionally oriented).

*We thank the reviewer for taking the time to review the manuscript and for their constructive feedback. The revised manuscript will include several changes based on the reviewer's comments which are outlined in more detail below under each specific comment.*

**Specific comments:**

- Despite the uplift due to the sloping isentropes of the front, there is little discussion of forcing by the upper levels. Presumably, the front is accompanied by a trough, and a simple geopotential contour or vertical motion could be useful for discussing the results.

*We initially planned to address forcing mechanisms in the next paper, however the climatological value of certain processes such as convergence, surface moisture and CAPE in the 700 hPa frontal framework would be useful for discussing the results. We will include a new figure (Figure R1; Figure 3 in the revised manuscript) that shows the climatological convergence at different pressure levels as a function of the distance from the 700 hPa front. This provides insight into the forcing at different levels. Thank you for the suggestion.*

- It is unclear why a characteristic such as cell lifetime or lightening has a local minimum at the location of the 700-hPa front. The authors are encouraged to comment more on this observation, which is clearly seen in Fig. 10 and only briefly discussed in l. 215. It seems to be a fortunate coincidence that the largest downward motion should occur at the location of the 700-hPa front, but the author should support their hypothesis for example, by plotting vertical motion, surface moisture, and CAPE/CIN.

*The new figure (Figure R1) mentioned in the comment above has the climatological CAPE, surface dewpoint and solar heating overlayed. We find that the CAPE is climatologically lowest surrounding the 700 hPa front. However, we believe a more comprehensive analysis would be required taking microphysics into consideration to unravel why the minimum lightning fraction is located at this distance relative to the front. Such an analysis is unfortunately beyond the scope of this current study as it is a climatology.*

[Figure]

*Figure R1: Climatological convergence at pressure levels (shaded), MUCAPE (dashed line), surface dewpoints (straight line), surface shortwave radiation (straight dotted line) in ERA5 depending on the distance from the 700 hPa front. Excluding convergence, the composite means are normalised between 0 and 1. The CAPE in ERA5 is calculated based on the most unstable parcel below 350 hPa.*

I think a map of the topography of Germany in steps of hundredths somewhere in the paper might help in understanding some of the local features, especially for the non-frontal cases.

*Thank you for the suggestion, Figure 8f in the revised manuscript with 500, 1000 and 1500 metres elevation and discusses how the spatial climatology relates to the orography (see L389–392 of tracked changes manuscript). Indeed, the highest frequency of both cold-frontal convective cells and non-cold-frontal convective cells can be found in the south of Germany in proximity to the Alps. The exception is post-700hPa-frontal cells which appear to be driven more by land-sea interactions.*

- Are the post-front cases (Fig. 7a) in the northwest related to land-sea circulations?

*This is an interesting question that we plan to address in a future study. We do briefly mention this in the current manuscript (see L378–381 of tracked changes manuscript). Testing this hypothesis would involve deeper analysis in ERA5 or observational data which we see as beyond the scope for this paper since we are primarily focusing on climatology.*

- Would be possible to add some information on the frontal strength in terms of temperature or humidity gradients alongside the characteristics of the convective cells (e.g., lifetime)? It looks like the 700-hPa front line is behind a strong gradient in humidity reminiscent of squall lines (Fig. 2a).

*This is also a very interesting question. We found that there is no clear relationship between the strength of the equivalent potential temperature gradient and the number of cells, i.e., a stronger equivalent potential temperature gradient at the front does not necessarily increase the number of cell detections. We suspect this highlights the importance of thermodynamic processes and other smaller-scale sources of lift on the development of convective cells, e.g., orography, outflow boundaries etc. We hope to address such aspects in more detail in a future study. We appreciate the reviewer's suggestion, but we reiterate that since the main focus is on climatology, we don't want to dive into such aspects too extensively in this paper.*

*Regarding Figure 2a the strong theta-e gradient ahead of the 700 hPa front is linked to a secondary (smaller-scale) cold-front that formed in the warm sector. See Figure R2 below.*

[Figure]

*Archived by www.wetter3.de*                                    *11-05-12 18 UTC*

*Figure R2: Synoptic analysis from the Met Office on 11th May 2012 at 18 UTC*

To justify the 750 km, the author could also perform a test against randomization. The number of additional cells attributed to a front when the distance is increased in 50-km increments can be plotted against the same but using radar data from a randomly chosen date. At the radius where both changes in the additional number of cells become equal, the increase in additional cells attributed to the front can no longer be distinguished from noise. Below this threshold, however, the increase is more than noise and is therefore physical.

*We randomly sampled cell-front distances by shuffling all cold front timesteps between 2007 to 2016 (April–September). This process was repeated 100 times, and the range of the samples is shown by the whiskers in the plot below indicating the 5th and 95th percentiles (Figure R3; Figure B1 in the revised manuscript). The mean of all 100 samples is represented by the orange line, while the blue line represents the cell count (real sample). We observe a sample bias for pre-700hPa-frontal cells. Instead of using the timestep count to assess the bias, we will include the figure below in the Appendix and refer the readers to it. In the warm-sector, the cell count is significantly higher than the randomised sample, with a maximum ratio of 2.8 (black line). The overlap between the random sample and the real sample is between 950–1100 km (95% confidence), with a second overlap at around 100 km. If we based our distance limits on the overlap of the randomised sample and the real sample, we would select 100 km and 1000 km. However, we also want to focus on the differences between pre-700hPa-frontal cells (more frequent than the random sample) and post-700hPa-frontal cells (less frequent than the random sample). As mentioned earlier, this figure will be included in the Appendix and referred to in the first results section. Thank you for the suggestion and the valuable addition to the paper.*

[Figure]

*Figure R3: Cell count (blue), random sample cell count mean (orange) and ratio of cell count and randomised sample (black) on the secondary axis. The random sample was performed 100 times and the 5th and 95th percentiles are shown by the whiskers of the boxplots.*

------------------------------------------------------------------------------------------------------------------------------------------ ----------

**Authors' responses in red italics**

**RC2**

**Review on "The climatology and nature of warm-season convective cells in cold-frontal environments over Germany" by Pacey et al.**

This study presents a climatology of convective cells associated with cold fronts in a front relative coordinate frame and compares these to convective cells occurring in non-cold frontal environments. Convective cells are shown to be much more frequent on cold front days than non- cold front days and the most likely location for convective cells to develop on cold front days is found to be 350 – 400 km ahead of the front. Overall, the manuscript is clear, well written and the results are supported by evidence. However, I have three major concerns regarding this manuscript.

*We thank the reviewer for taking the time to review the manuscript and for their constructive feedback. The revised manuscript will include several changes based on the*

*reviewer's comments which are outlined in more detail below under each specific comment.*

The first major comment concerns how well this study fits the scope of this journal and the broader context of the results. In the manuscript the link to actual hazards is weak and little emphasis is given to this aspect. Lightning is considered but relatively briefly. It should be clearer how the results of this study inform about meteorological hazards. This study also only focuses on Germany, a choice which is motivated by the availability of radar data. While the authors do state that the study should be expanded to all of Europe, the current manuscript may be of limited interest to readers from other places than Germany. At a minimum the authors should attempt to address the question of how do these results apply to elsewhere in the world? Do they only apply over continental areas for example?

*We thank the reviewer for their feedback regarding the suitability of the journal and applications to other parts of Europe. We do think this study is a good fit for the journal, since it focuses on convective cells (defined by areas exceeding 46 dBZ), which are known to be linked to hazards such as lightning, wind, rain and hail. Indeed, from Figure 10d we see that most cells are indeed associated with lightning; around 80% of cells in the warm-sector, which is a higher percentage than non-cold-frontal cells. Furthermore, in the final section of the paper (Section 3.4) we discuss in detail regarding the links to hazards e.g., intensity of cells and mesocyclone frequency in the cold-frontal framework. For example, we look at the fraction of cells with large 55 dBZ cores, a commonly used threshold for hail. So that it is clearer that the study furthers understanding of convective hazard climatology, we have added an additional line in the abstract that highlights a key result regarding mesocyclones. Furthermore, additional discussion has been added to the introduction to highlight that the motivation of this work relates to convective hazards (see L64–65 of tracked changes manuscript).*

*Regarding the application to other parts of Europe we found that lightning frequency has a similar distribution around the 700 hPa front in a larger European domain (see Figure R1; Figure A2 in the revised manuscript). This gives us an early indication that the results could apply on a broader European scale but since radar data (especially doppler wind velocities which are needed for mesocyclone detection) is not readily available on a European scale it is not feasible to reproduce all the results on a European scale at this time. We have included some discussion regarding our preliminary results on lightning frequency in a larger European domain in the new manuscript (see L316–318 of tracked changes manuscript).*

[Figure]

*Figure R1: Lightning strike frequency for a sub-European domain (see grey domain in Figure 1) depending on the distance from the 700 hPa front between 2010–2016 (April–September). Lightning data were provided by the Met Office, which uses an arrival time difference network (ATDnet) to detect lightning strokes.*

The second major comment regards the assumption that the surface front is 300 km ahead of the 700-hPa front. This is an oversimplification and likely is not accurate in many cases. Specific points:

1. It is stated that this assumption is based on ERA5 data, but this is not presented – it should certainly be shown even if only as supporting material.
2. It is stated that the surface convergence zone is 300 km ahead of the 700-hPa front. In some cold frontal cases the wind shift (i.e. convergence zone) is not co-located with the temperature gradient so it may not be the thermal gradient which is 300 km ahead of the 700-hPa front. It is also inconsistent to use convergence to locate the surface front but a thermal gradient for the 700-hPa front.
3. This simple approach does not consider kata-cold fronts in which the front appears to slope forward with height as the cold air aloft has overrun the surface front. Kata cold fronts can certainly trigger elevated convection and these fronts should be considered separately.

*We thank the reviewer for their feedback regarding the surface front location relative to the 700 hPa front. The assumption is not only based on the maximum climatological near-surface convergence in ERA5 but also on knowledge of cold frontal slopes (1:100). We do not claim the surface front is always located exactly 300 km ahead of the 700 hPa front rather this is where the mean location of the surface front is. For this paper we are primarily interested in cold-frontal convective cell climatology. We are not focusing on specific case studies where, as the reviewer rightly mentions, the surface front location relative to the 700 hPa front may vary. We have added a new subsection (3.1.5) which states our key assumptions regarding the surface front location relative to the 700 hPa front.*

1. *The climatological convergence at different height levels in the 700 hPa framework is included in the revised manuscript as the reviewer suggests (see Figure R2; Figure 3 in the revised manuscript). The climatological value of CAPE, surface dewpoints and surface shortwave radiation are also overlayed.*

[Figure]

*Figure R2: Climatological convergence at pressure levels (shaded), MUCAPE (dashed line), surface dewpoints (straight line), surface shortwave radiation (straight dotted line) in ERA5 depending on the distance from the 700 hPa front. Excluding convergence, the climatological means are normalised between 0 and 1. The CAPE in ERA5 is calculated based on the most unstable parcel below 350 hPa.*

2. *Since near-surface convergence is relevant for convective initiation, we believe this is an appropriate reference point. Furthermore, the surface temperature gradient is typically not as well-defined, especially in the warm-season. For this reason, automatic front detection methods are usually applied above the boundary layer.*

3. *The thermodynamic gradient of kata fronts still backs with height as with an ana front (Moore and Smith, 1989; their Figure 1), thus the approach holds for both types of fronts. Furthermore, if these upper-level humidity fronts were being detected using our cold-front detection methods we would expect to find a much larger cell frequency behind the 700 hPa front. Since Kata fronts typically evolve from Ana fronts, it is not trivial to separate such fronts on a case-by-case basis. It may however be interesting for future work to investigate the effects of the overrunning upper-level dry intrusion on the resulting convection when a kata cold front is present. It is expected this increases the strength of the capping inversion thus inhibiting the premature release of convection which is relevant for severe convective storms. In the revised manuscript we have noted that we do not explicitly consider ana and kata cold fronts separately (see L206–207 of tracked changes manuscript).*

The third major comment concerns the clustering presented in section 3.3.3. Specific issues here are:

1. The manuscript lacks details on exactly how the clustering was done (e.g. was any normalisation on the input features performed?).
2. The choice of the number of clusters appears subjective whereas the silhouette score and elbow plots could be used to better justify the final number of clusters.
3. The number of clusters (30) is too large to be of practical use to e.g., forecasters
4. The justification for removing 6 of these 30 clusters is not clear and it appears that the clustering has identified 6 clusters which are not physically consistent – strongly suggesting that the clustering has not been performed in an optimal manner.
5. The only outcome of the clustering that is presented is the number of cells and the location of the front. It would be helpful for forecasters to also see additional meteorological variables associated with each of these clusters, for example, the MSLP and equivalent potential temperature.

1. *We thank the reviewer for their feedback on the clustering. We agree that further details would be useful in this section. For the reviewer's information, no normalisation was carried out on the input data as it is binary, each grid point is either 1 for a front grid point or 0 for a non-front grid point.*

*2. The reviewer is right that the elbow method or silhouette score can be used to select the cluster number in a more objective manner. However, no clear elbow was identified using the elbow method and no optimal silhouette score (see Figure R3; Figure C1 in the revised manuscript). In this case user expertise was used instead of the metrics. Further justification of the cluster number choice is discussed in the responses below. Please see additional discussion in the revised manuscript (see L405–420 of tracked changes manuscript).*

[Figure]

*Figure R3: Elbow method (left) and silhouette score method (right) applied for cluster numbers between 2 and 50.*

*3. We do not think that the cluster size is too large, forecasters are aware that cold fronts have a variety of orientations and different geographical locations. For example, forecasters at the German Weather Service (DWD) are used to identifying 29 different synoptic weather patterns (Wapler and James, 2014, their table 1). We carried out extensive testing between 10 and 50 clusters finding that lower cluster numbers resulted in fronts with different features being grouped in the same cluster. Larger cluster numbers on the other hand had less variance in each cluster but the results become less interpretable. Ultimately, a compromise somewhere must be found.*

*4. 6 clusters were removed due to high variance within those clusters. This however does not indicate the clustering has not been performed in an optimal manner. When using a higher cluster number such as 50 these 6 clusters would have been split into separate*

*clusters with less variance in the cluster, but as mentioned at 50 clusters it is hard for the reader to interpret such results.*

*5. The primary focus of this analysis is to see how the orientation/position of the front is linked to the cell detections. A composite of the equivalent potential temperature field for each cluster will only show a boundary where the front is located, thus we don't feel this will add any additional information.*

Below I also list some minor comments which would certainly improve the manuscript:

Minor comments

1.  Line 41 – 42. Please expand this sentence to make it clearer. It needs to be stated that this is due to the frontal surface sloping rearwards with height. *On L39 we already mention the cold-front slope, this has been amended to read "Since cold fronts typically slope backwards with height, there is no concrete...". Thank you for the suggestion (see L43 of tracked changes manuscript).*
2.  Line 62, Question 3. This could be written in a manner so it can stand alone and does not need a reader to refer back to Q1 and Q2. This would likely make it clearer and easier to understand. *We think this would lead to a rather long research question. The reader would have just read questions 1 and 2 so it will be clear what is meant by question 3, we prefer to avoid unnecessary repetition in this case.*
3.  Line 125. Figure 1. Could the domain where fronts are identified in be marked on this figure? *The domain showed in Figure 1 is the domain in which fronts were detected. We have noted this in the Figure 1 caption in the revised manuscript. Thank you for the suggestion.*
4.  Section 2.1.1. The criteria used to identify the fronts are quite large so will only identify quite strong fronts in terms of the thermal gradient. Do the results depend on these thresholds, and in particular, do the fronts still hold if weaker fronts are also considered? If not, it should be stressed more clearly that these results only apply to strong cold fronts. *We have emphasised in the method section (section 2.3) that we are focusing on cases where a single large-scale cold front is present in western/central Europe, and that smaller-scale and weaker fronts (below 6K / 100km) are not explicitly considered (see L203–206 of tracked changes manuscript). Thank you for the comment.*
5.  Section 2.3. How were these four examples selected and how representative of the whole data set are they? They look like quite standard fronts, so I am wondering if the method works well with more complicated or less uniform fronts. *The examples were selected to show fronts with different orientations and*

*with cells in different locations relative to the front. The clustering analysis shows the primary frontal types contained in the dataset. The study does indeed focus on standard fronts where only one large-scale cold front is present over western/central Europe.*

6. Section 2.3 / method. "Timesteps containing two or more cold front lines in the domain were omitted". Since timesteps with two fronts present are omitted, this means that this method only works for a small area and could not be expanded to e.g., European scale (even if the radar data was available). This is a notable limitation of this method which should be clearly highlighted, or the method improved to allow two or more fronts to be present at the same timestep. *In the event there are two large-scale fronts (L~1000 km) present it is not trivial to decide which front to use for the cell-distance calculations. Selecting the largest front for example may induce a bias as the smaller front may also have some influence on the resulting convection. As mentioned in the last comment we will remark in the revised manuscript that we focus on cases with a single large-scale (L~1000 km) cold front present over western/central Europe.*

7. Related to the point above, neglecting timesteps with two or more fronts present means that double fronts will be automatically ignored which may add a systematic bias to the results. *We believe the opposite is true as when two large-scale fronts are present it is not clear which front should be selected without inducing a systematic bias.*

8. Line 176. There is a typo here "in In Figure 2a…" *Thank you, this has been corrected.*

9. Section 3.1. There are a lot of numbers, percentages especially at the start of this section and it is hard to read. Many of these numbers etc, are in Table 1, but Table 1 is not referred to much here. It would help a reader to refer to Table 1 more. Furthermore, this section may be clearer if the number of cold front days was discussed first, and this information was added to table 1. *We agree that it would be useful to refer to Table 1 earlier in the section, thank you for the suggestion. The second line has been removed as the information is already contained in the table, thus making the text shorter.*

10. Line 199, should this be a comma before All rather than a period? *We feel including this in the same sentence would make it too lengthy, however the sentence could be rephrased for clarity. We have revised it to read "Such synoptic patterns are likely…" (see L287–288 of tracked changes manuscript).*

11. Line 210. Is the surface convergence influenced by land use, coastlines, topography etc.? *ERA5 at ~25 km resolution may struggle to resolve such features and*

*when averaging across several thousand timesteps such effects will be filtered out leaving the surface convergence due to the front.*

12.Line 218 and elsewhere after this the phrase "pre-700-frontal environment..." is used*. Pre-700hPa-frontal and post-700hPa-frontal are used throughout the revised manuscript for consistency, thank you for bringing this to our attention.*

Should hPa be added after 700 here? *Pre-700hPa-frontal and post-700hPa-frontal are used throughout in the revised manuscript.*

13.Line 272 – 274. How does this spatial climatology of convective cells relate to the spatial climatology of fronts as shown in Figure 1? Adding a sentence to relate these aspects would be helpful for a reader. *North-west Germany has the largest fraction of cell days on cold-frontal cell days which could indeed be linked to the high frequency of fronts seen in this region. We have mentioned this in the section 3.3.2 of the revised manuscript (see L395–396 of the tracked changes manuscript). Thank you for the comment.*

*References*

*Moore, J. T., and K. F. Smith, 1989: Diagnosis of Anafronts and Katafronts. Wea. Forecasting,* **4***, 61–72, https://doi.org/10.1175/1520-0434(1989)004<0061:DOAAK>2.0.CO;2.*

*Met Office, 2020: Lightning strike location data. Met Office, 3 pp., https://www.metoffice.gov.uk/binaries/content/assets/metofficegovuk/pdf/data/adtnet_data_sheet.pdf.*

*Wapler, K. and James, P, 2014: Thunderstorm occurrence and characteristics in Central Europe under different synoptic conditions, Atmospheric Research, 158, https://doi.org/10.1016/j.atmosres.2014.07.011.*
* * *
***Authors' responses in red italics***

**RC3**: 'Comment on nhess-2023-39', Anonymous Referee #3, 05 May 2023  reply

**General comments:**

This study analyses the location and characteristics of deep moist convection associated with cold fronts over central Europe. Many novel insights are gained and nicely embedded in the existing literature. Overall, the methods, structure, and the figures are of a high quality. I don't have any reasons for rejection and my comments can mostly be seen as suggestions, although I agree with some of the concerns of reviewer #2 (see major comment 3 below).

*We thank the reviewer for taking the time to review the manuscript and for their constructive feedback. The revised manuscript will include several changes based on the reviewer's comments which are outlined in more detail below under each specific comment.*

**Major comments:**

1) line 132: The bias in Fig. 1 looks more than "slight". If it were a weak bias, shouldn't more fronts be expected towards the Atlantic where strong lows are originating from? Or is the theta gradient increasing over land? I think some more discussion for why the dataset is still useful for your purpose seems warranted, e.g., that you are mostly interested in fronts with convection impacting central Europe.

*We have revised the manuscript to indicate that there is a bias towards the NW and SE of the domain and since we focus on Germany this does not bias our results (see L148–149 of the tracked changes manuscript). Thank you for the suggestion. However, we note that several other front climatologies also did not find the highest front frequency in the Northern Atlantic e.g., Fig. 5c in Schemm et al. (2015) and Figure 7 in Niebler et al. (2022).*

2) 159 Do you think any biases resulted from only counting the first cell detection? I think the approach is good enough, but I could imagine that long-lived cells change their location relative to the front over time?

*Figures 4 and 5 in the current manuscript were also created using cells at all detection times and no significant differences were observed. From Figure 10a and Figure 10b in the current manuscript we see that cells have a typical lifetime between 15-20 minutes and propagate at a speed of between 45–65 km hr$^{-1}$ in cold frontal environments. The mean distance a cell propagates is therefore between 11–21km. We produce the results in 50 km intervals, supporting why we see no significant differences by including cells at later detection times. The primary motivation for only counting the first detection was to remove duplicate counting of cells.*

3) Perhaps my main point of criticism (or the aspect of the study which could be improved the most) is that although you discuss lift mechanisms ahead and at the front, not much analysis is done to locate these features relative to the 700hPa front location. I realize that this is not easy and changes a lot from case to case, but since the

study claims to describe the "nature of cold-frontal convection", a deeper analysis seems warranted. For instance, is it possible to add the locations of the surface front and pre-frontal convergence relative to the 700hPa front in Figs. 4 and 5 (or rather the range of locations in your dataset, e.g., as a box and whisker)? Could this be estimated from the ERA5 data you used? An alternative would be to go through some cases manually and indicate these boundary positions for each case in the plots.

*We thank the reviewer for their feedback. Figure R1 below (Figure 3 in the revised manuscript) shows the climatological convergence at different pressure levels as a function of the distance from the 700 hPa front. This shows the typical forcing from the front at different distances from the front and at different height levels. This was originally planned to be left for a future publication, but we agree this analysis would be useful for readers. The climatological value of CAPE, surface dewpoint and solar radiation are also overlayed and we discuss how the results from the climatology relate to Figure 3.*

[Figure]

*Figure R1: Climatological convergence at pressure levels (shaded), MUCAPE (dashed line), surface dewpoints (straight line), surface shortwave radiation (straight dotted line) in ERA5 depending on the distance from the 700 hPa front. Excluding convergence, the climatological means are normalised between 0 and 1. The CAPE in ERA5 is calculated based on the most unstable parcel below 350 hPa.*

4) 295 Looking at the number of cells might be misleading because larger storm systems (MCS) result in less cells counted even though they have a larger impact. You don't need to change the analysis but this should be made clear again to the reader. In

general, strengths and weaknesses of the KONRAD dataset are not discussed much.

Another example would be that you speculate that the pre-frontal diurnal cycle is broader because of more MCS. Wouldn't that also mean that less individual cells were detected there (because one MCS has larger but less cells)? The opposite is seen in Fig. 10c. Is that because of flaws in the detection algorithm?

*We have seen from a few case study examples that MCS's are typically associated with a larger number of cells. There are two explanations for this, firstly a continuous line exceeding 46 dBZ may not always be present, therefore several cells are detected within an MCS. Secondly, due to cell recycling in an MCS, KONRAD will likely detect new cells at subsequent timesteps. See an example case below (Figure R2) with 4 radar timesteps shown (15-minute interval) and KONRAD cell detections at 5-minute intervals (bottom) for a case study in August 2013. We will not include these figures in the manuscript but we have added discussion of the llimitations of KONRAD  (see L181–189 of the tracked changes manuscript).*

[Figure]

[Figure]

*Figure R2: OPERA Radar Data showing max dBZ (Huuskonen et al. 2014; Saltikoff et al. 2019) (top). KONRAD cell detections with first detections in blue and secondary (or later) detections in orange (bottom).*

**Minor comments and technical corrections:**

24-25 An alternative to this explanation would be varying DMC ingredients (e.g., CAPE) in different regions along the front. *We have revised the manuscript according "Furthermore, the importance of difference mechanisms may vary across the front, e.g., stronger synoptic lift near the front but increases in solar heating away from the front." (see L27–29 of the tracked changes manuscript). Thank you for the suggestion. The varying DMC ingredients across the front is also now shown in the Figure 3.*

26-38 There is a lot of good content here, but the text seems a bit unstructured. For instance, pre-frontal convergence lines are also a result of an ageostrophic circulation (Dahl and Fischer 2016), not only the lift at the front. Also, you could make a bit clearer that you start discussing mechanisms ahead of the front, then at the front and then behind, for instance by ending the first sentence (l. 27) with "... location relative to the front." And then starting the next sentence with "Ahead of the front, ...". *We have reorganised the sentences in the introduction section based on the reviewer's feedback (see L30–37 of the tracked changes manuscript). Thank you for the suggestion.*

41-42 I'm not quite sure if I understand the point. If this holds true for a surface observer, why is it not true? Are you referring to the fact that the convective cloud is not necessarily in the same location as the heaviest precipitation? *We are pointing out that if convection is observed shortly after the passage of the surface front, the saturated cloud region is largely on the warm-side of the front thus is not post-frontal. Indeed, the precipitation falls to the ground into the cooler post-frontal airmass, but the cell itself is typically on the warm-side of the front.*

97-98 If I understand correctly, "higher" would only be true for cold fronts, because for warm fronts $v_f$ would be negative. Did you mean to say the magnitude of $v_f$ is higher with stronger advection? *Yes, thank you. The manuscript has been revised accordingly (see L107–108 of the tracked changes manuscript).*

101-104 Remains unclear how L was determined. Is it a continuous length of points where the other criteria were met? What about brief gaps in the boundary detection? *The coordinates of where TFP=0 are located using interpolation. The distance from each adjacent point was calculated and summed for the whole line. In the event there is a gap this is considered a separate feature and not added to the total length. We have expanded on our current explanation to include the information mentioned above (see L118–119 of the tracked changes manuscript).*

141 "At the start of this study," (comma in English after such introductory words for a sentence" *Thank you.*

section 2.2 in general: Just a suggestion, but I would bring in the hail, lightning, and mesocyclone detection methods later when they are needed. Jumping back and forth between the different dBZ thresholds was a bit confusing here. *We did consider this, but we felt it would disrupt the flow of the results to include such technical definitions in the results section. We have instead included a new table (Table 2) with the different thresholds, so it is easier for the reader refer back to. Thank you for bringing this to our attention.*

155 Make clear what "such" is referring to. I'm assuming you mean the hail, lightning and mesocyclone detections? *Thanks for the suggestion, indeed it is referring to the hail flag, lightning and mesocyclone definitions. The manuscript has been revised accordingly on L173–175 of the tracked changes manuscript.*

163 comma behind "2.2". *Thank you.*

164 I like this numbering of criteria. Makes it really clear. *Thank you for the feedback.*

185 Comma after "September" *Thank you, the manuscript has been revised accordingly.*

187 and 262 One convective cell is a fairly low threshold. Days with >1 and >100 days are weighted equally with this method, correct? Perhaps discuss this caveat of considering cell days by e.g., showing a histogram of the number of cells per cell day to make clear that most cell days were really days with much convective activity?

*The one cell per day criterion is just to assign a day as a cold-frontal or non-cold-frontal cell day. For this analysis we are actually interested in the mean number of cells per day so by just selecting days with say >20 cells would add a bias to the results.*

189 Associated "with" *Thank you.*

239-240 There is a clear secondary maximum around 750 km ahead of the front in Fig. 5, which is interesting. Do you speculate that this is just free convection in the warm sector or pre-frontal convergence serving as another (weak) trigger? By the way, I really like this Figure. Is it necessary to have non-linear color scheme. It may look nicer but I think it makes it harder to interpret the numbers. If it's necessary, it should at least be mentioned in the caption. *From Figure 4 we see that between 600–750 km the cell frequency remains stable with a slight increase. It is plausible that this is linked to pre-surface-frontal convergence lines as they are usually found around 300km ahead of the surface front. Additional discussion has been added to the revised manuscript (see L303–304 of the tracked changes manuscript). Thank you for bringing this to our attention.*

*Since there are a lot more cells pre-700hPa-frontal compared to post-700hPa-frontal it is necessary to use a non-linear colorbar to highlight the post-700hPa-frontal diurnal cycle. This has been noted in the figure caption, thank you for the suggestion.*

248-255 How do supercells fit into this picture? Their long lifetime could also lead to a broader diurnal cycle. You only mention Wapler (2016) briefly and without context. *We have added a line after the Wapler (2016) reference that says "This indicates that supercells, which typically have a longer lifetime, may also be linked to the observed weakened diurnal cycle" (see L359–360 of the tracked changes manuscript). Thank you for the suggestion.*

259 First time reading this, I was confused what you mean by "vary". It might just be me, but if you refer to the spatial distribution, it's clearer to say something like: "The frequency of convective events varies in different parts of Europe." *Thank you for the suggestion, this will indeed make it clearer to readers. This has been revised accordingly.*

268-270 Sounds like you believe this is due to a general increase towards the south / the mountains. Your resolution is fairly coarse, but the spatial distribution you observe would also be consistent with mesoscale enhanced convective activity in local parts of Germany, e.g., South of Stuttgart (Piper 2017 Fig. 3, Kunz 2010). *Thank you for the reference, this has been added to the revised manuscript.*

295 Here and elsewhere: "Colorbar" *Thank you.*

303 This is consistent with enhanced activity in the Erzgebirge region (Piper Kunz 2017). *Thank you for the reference, this has been added to the revised manuscript.*

322 Such a pattern is also often associated with advection of an elevated mixed layer from Southwest Europe and pre-frontal convergence lines (Dahl and Fischer 2016). *Thank you for suggestion and reference, this has been added to the revised manuscript (see L452–455 of the tracked changes manuscript).*

378-385 This last paragraph was a nice finish and the results are plausible. The conclusion section is also nice and precise. *Thank you for the feedback.*

Fig. 8 I also liked this Figure and analysis. Could you add the average number of cells per event over whole domain in the top of each subplot? Even though some clusters might be rare, their impact/number of cells might be large, so giving the reader information about the typical number of cells could be useful. *The number of cells per grid box is already normalised by the number of timesteps in the cluster accounting for rare cluster types.*

Fig. 10 Also very informative plot. Titles for each subplot would be helpful to avoid having to jump between caption and plot. *After reviewing the figure again, we think the labels on the y-axis are sufficient. The reader does not need to jump between the figure and the caption since they can refer to the y-axis labels.*

*References*

*Huuskonen, A., E. Saltikoff, and I. Holleman, 2014: The operational weather radar network in Europe. Bull. Amer. Meteor. Soc., **95**, 897–907, https://doi.org/10.1175/BAMS-D-12-00216.1.*

*Schemm, S., Rudeva, I., and Simmonds, I.: Extratropical fronts in the lower troposphere–global perspectives obtained from two automated methods, 2015: Quarterly Journal of the Royal Meteorological Society, 141, 1686–1698, https://doi.org/10.1002/qj.2471*

*Saltikoff, E., and Coauthors, 2019: OPERA the Radar Project. Atmosphere, **10**, 320, https://doi.org/10.3390/atmos10060320.*

*S. Niebler, A. Miltenberger, B. Schmidt, and P. Spichtinger, 2022: Automated detection and classification of synoptic-scale fronts from atmospheric data grids. Weather and Climate Dynamics, 3(1):113–137. doi: 10.5194/wcd-3-113-2022. URL https://wcd.copernicus.org/articles/3/113/2022/*

---

## Referee Report (RR1)

**Review on "The climatology and nature of warm-season convective cells in cold-frontal environments over Germany" by Pacey et al.**

I would like to thank the authors for considering all of my previous comments and revising the manuscript in an appropriate way. I especially appreciate the addition of Figure 3 which considerably strengthens the manuscript. However, in my opinion the authors could still improve the manuscript regarding two of my previous major comments:

Previously I wrote *"The first major comment concerns how well this study fits the scope of this journal and the broader context of the results. In the manuscript the link to actual hazards is weak and little emphasis is given to this aspect. Lightning is considered but relatively briefly. It should be clearer how the results of this study inform about meteorological hazards."* Although this has been improved I still find the link from the convective cell characteristics to hazards weaker than it could be. Some specific comments related to this:

1. Line 64. Mesocyclones are rather suddenly mentioned at the end of the introduction with little background given to why the presence of these features would lead to more hazardous / extreme weather. Details could be added to the introduction on how does the presence of a mesocyclone related to a hazard.

2. Do the hazards (lightning, hail) associated with convective cells vary if they are on the pre-frontal or post-frontal side of the cold front? This is some what included in question 2 in the introduction and it is in the analysis but it is not clearly stated in the introduction that this is covered in the manuscript. Another place where the link between meteorological features and hazards could be strengthened is on lines 78 – 79 where it is stated that "For the nature of cells we investigate cell lifetime, propagation speed, organisation, lightning frequency, cell intensity, and mesocyclone frequency" → here text could be added explicitly stating that how hazards vary by distance from the front are investigated.

3. In the response the authors state *"We will also emphasise in the conclusion that this work improves understanding of convective hazard climatology"* but when reading the revised conclusions I see that details concerning the results from the new Figure 3 have been added but text about how hazards (hail, lightning) relate to fronts as identified from this study is still mainly lacking.

The second major comment that I feel the authors could do more to address regards the clustering. Additional details about the clustering have been added, which I appreciate, but I still feel the justification for using k=30 **then** removing 6 clusters is weak. In particular, I find it hard to understand why this is an more appropriate choice than using k=24. At a minimum the authors should show the 6 clusters that the remove from their analysis. Furthermore, Figure 9 could be reproduced in the supplementary material with a few different choices of the number of clusters so that a reader can see how sensitive the results are. In particular, there is a localised maximum in the Silhouette score at k=9 so this would be interesting to present – and if the results do not show something physically meaningful this would actually strengthen the authors choice of k=30.

Minor comments:
1. The caption in Figure 3 could be clearer regarding the description of the lines. Suggest using "...CAPE (dashed line), surface dewpoints (solid line), surface shortwave radiation (solid line with circular markers)".

2. Line 226-227. The addition of Figure 3 makes many of the conclusions presented in this manuscript much more robust and I'm really pleased to see some evidence to support the commonly written claim that cold fronts have a slope of 1:100 – thank you. However, how

exactly has figure 3 been created? Does every front / convective cell pair contribute values at all grid points shown in this figure domain? e.g. for each front is the cross section extracted from ERA5 and then all of these averaged? Can a few additional details be added here? Furthermore, it is not clear how or why the normalisation has been done for CAPE, dewpoint temperature and solar radiation. Can these details also be added.

3. Section 3.1.2 / CAPE. What type of CAPE is this? Most unstable CAPE? Surface CAPE?
4. Line 255. Suggest to move "(straight dotted line in Figure 3) earlier in this sentence as it currently implies that cloud cover is plotted in Figure 3. Also see minor comment #1 above regarding line description.

5. Line 264 – 266. While I think it is now fine to state that the surface front is on average 300km ahead of the 700-hPa front (e.g. there is evidence for this in Figure 3), the authors still assume that all fronts are the same. This assumption could be supported by computing some estimate of variability in the parameters shown in Figure 3. e.g. what is the standard deviation of the convergence or the 25-75% percentiles of the CAPE values (could be shown on Figure 3). While this is not essential, it would strengthen the manuscript.

---

## Author Response (AR2)

*Authors' responses in red italics*

Submitted on 05 Sep 2023
Anonymous referee #1

*We thank the reviewer for taking the time to review the manuscript again and for their constructive comments and feedback in the previous round.*
* * *
Submitted on 28 Sep 2023
Anonymous referee #2

**Review on "The climatology and nature of warm-season convective cells in cold-frontal environments over Germany" by Pacey et al.**

I would like to thank the authors for considering all of my previous comments and revising the manuscript in an appropriate way. I especially appreciate the addition of Figure 3 which considerably strengthens the manuscript. However, in my opinion the authors could still improve the manuscript regarding two of my previous major comments:

*We thank the reviewer for reading through the revised manuscript and providing further constructive feedback. We provide a point-by-point response below.*

Previously I wrote *"The first major comment concerns how well this study fits the scope of this journal and the broader context of the results. In the manuscript the link to actual hazards is weak and little emphasis is given to this aspect. Lightning is considered but relatively briefly. It should be clearer how the results of this study inform about meteorological hazards."* Although this has been improved I still find the link from the convective cell characteristics to hazards weaker than it could be. Some specific comments related to this:

1. Line 64. Mesocyclones are rather suddenly mentioned at the end of the introduction with little background given to why the presence of these features would lead to more hazardous / extreme weather. Details could be added to the introduction on how does the presence of a mesocyclone related to a hazard.

*We have added additional text introducing mesocyclones and their link to hazards (see L57–60 of tracked changes manuscript). Thank you for the suggestion.*

2. Do the hazards (lightning, hail) associated with convective cells vary if they are on the pre- frontal or post-frontal side of the cold front? This is some what included in question 2 in the introduction and it is in the analysis but it is not clearly stated in the introduction that this is covered in the manuscript. Another place where the link between meteorological features and hazards could be strengthened is on lines 78 – 79 where it is stated that "For the nature of cells we investigate cell lifetime, propagation speed, organisation, lightning frequency, cell intensity, and mesocyclone frequency" →here text could be added explicitly stating that how hazards vary by distance from the front are investigated.

*We have added the following line after the quoted line. "The nature of cells section will therefore provide novel insights into how convective hazard climatology varies depending on the distance from the front." (see L79–80 of tracked changes manuscript) Thank you for the suggestion.*

3. In the response the authors state "*We will also emphasise in the conclusion that this work improves understanding of convective hazard climatology*" but when reading the revised conclusions I see that details concerning the results from the new Figure 3 have been added but text about how hazards (hail, lightning) relate to fronts as identified from this study is still mainly lacking.

*We feel the link to hazards is already covered in the conclusions. The results using the 55 dBZ threshold is mentioned several times as well as lightning, and mesocyclones. We remind the reviewer that convective cells (defined by 46 dBZ) are a hazard themselves, the corresponding rain rate could cause a flooding hazard. Furthermore, the forecasting of hail for example is not possible without first identifying where the convection will initiate in the first place.*

The second major comment that I feel the authors could do more to address regards the clustering. Additional details about the clustering have been added, which I appreciate, but I still feel the justification for using k=30 **then** removing 6 clusters is weak. In particular, I find it hard to understand why this is an more appropriate choice than using k=24. At a minimum the authors should show the 6 clusters that the remove from their analysis. Furthermore, Figure 9 could be reproduced in the supplementary material with a few different choices of the number of clusters so that a reader can see how sensitive the results are. In particular, there is a localised maximum in the Silhouette score at k=9 so this would be interesting to present – and if the results do not show something physically meaningful this would actually strengthen the authors choice of k=30.

*The goal of k-means clustering is to separate similar data into clusters by minimising within-cluster variance and maximising between-cluster variance. Minimising within-cluster variance can usually be achieved by increasing the cluster number. However, the results become less interpretable and there could also be more clusters with similar features (low between-cluster variance). Lowering the cluster number could increase the between-cluster variance at the expense of increasing within-cluster variance. Running the algorithm with 30 clusters then removing 6 clusters seems to provide a good balance between low within-cluster variance and high between-cluster variance for the remaining 24 clusters. The 6 removed clusters have high within-cluster variance, i.e., they contain fronts with different shapes and orientations, so we don't see any reason to include this as supplementary material. We have however produced plots for cluster numbers 15, 35 and 50 and put them in appendix as the reviewer suggests.*

*In summary, there is of course always potential for further optimisations with such machine learning algorithms, but one must also be pragmatic. We hope this settles the reviewer's concerns.*

Minor comments:

1. The caption in Figure 3 could be clearer regarding the description of the lines. Suggest using

   "...CAPE (dashed line), surface dewpoints (solid line), surface shortwave radiation (solid line with circular markers)".

   *Thank you for the suggestion, we have amended the manuscript.*

2. Line 226-227. The addition of Figure 3 makes many of the conclusions presented in this manuscript much more robust and I'm really pleased to see some evidence to support the commonly written claim that cold fronts have a slope of 1:100 – thank you. However, how

exactly has figure 3 been created? Does every front / convective cell pair contribute values at all grid points shown in this figure domain? e.g. for each front is the cross section extracted from ERA5 and then all of these averaged? Can a few additional details be added here? Furthermore, it is not clear how or why the normalisation has been done for CAPE, dewpoint temperature and solar radiation. Can these details also be added.

*The means are calculated using all instances that an ERA5 grid point has a certain front distance and is not weighted by individual timesteps. We have added this line to the revised manuscript.*

*The variables are normalised to compare the typical magnitude of each variable at different distances from the front. This was already briefly mentioned, but we have now broken it down to a separate sentence so that it is clearer (see discussion between L219–225 of tracked changes manuscript).*

*Thank you for the suggestions.*

3. Section 3.1.2 / CAPE. What type of CAPE is this? Most unstable CAPE? Surface CAPE?

*Most unstable CAPE. This is mentioned in the last line of the Figure 3 caption.*

4. Line 255. Suggest to move "(straight dotted line in Figure 3) earlier in this sentence as it currently implies that cloud cover is plotted in Figure 3. Also see minor comment #1 above regarding line description.

*Amended. Thank you.*

5. Line 264 – 266. While I think it is now fine to state that the surface front is on average 300km ahead of the 700-hPa front (e.g. there is evidence for this in Figure 3), the authors still assume that all fronts are the same. This assumption could be supported by computing some estimate of variability in the parameters shown in Figure 3. e.g. what is the standard deviation of the convergence or the 25-75% percentiles of the CAPE values (could be shown on Figure 3). While this is not essential, it would strengthen the manuscript.

*We appreciate the suggestion, but we feel adding additional lines would make the plot too crowded.*
* * *
Submitted on 19 Sep 2023
Anonymous referee #3

The authors have thoroughly addressed my comments and I did not find anything else to improve. Great work!

*We thank the reviewer for taking the time to review the manuscript again and for their constructive comments and feedback.*

One minor comment is below:

line 220: Perhaps add a reference supporting your point that this is a typical slope for cold fronts?

*We have added the following reference. Thank you for the suggestion.*

*Bott, A.: Synoptische Meteorologie: Methoden der Wetteranalyse und -prognose, Fronten und Frontalzonen, p. 397, Springer Berlin Heidelberg, Berlin, Heidelberg, https://doi.org/10.1007/978-3-662-48195-0_11, 2016b*